# Cytochrome *c* lysine acetylation regulates cellular respiration and cell death in ischemic skeletal muscle

Paul T. Morse [1,9], Gonzalo Pérez-Mejías [2,9], Junmei Wan[1], Alice A. Turner[1,3], Inmaculada Márquez [2], Hasini A. Kalpage[1], Asmita Vaishnav[3], Matthew P. Zurek[1,3], Philipp P. Huettemann[1], Katherine Kim[1], Tasnim Arroum [1], Miguel A. De la Rosa [2], Dipanwita Dutta Chowdhury[3], Icksoo Lee[4], Joseph S. Brunzelle[5], Thomas H. Sanderson [6], Moh H. Malek[7], David Meierhofer[8], Brian F. P. Edwards [3], Irene Díaz-Moreno [2] ✉ & Maik Hüttemann [1,3] ✉

Skeletal muscle is more resilient to ischemia-reperfusion injury than other organs. Tissue specific post-translational modifications of cytochrome *c* (Cyt*c*) are involved in ischemia-reperfusion injury by regulating mitochondrial respiration and apoptosis. Here, we describe an acetylation site of Cyt*c*, lysine 39 (K39), which was mapped in ischemic porcine skeletal muscle and removed by sirtuin5 in vitro. Using purified protein and cellular double knockout models, we show that K39 acetylation and acetylmimetic K39Q replacement increases cytochrome *c* oxidase (COX) activity and ROS scavenging while inhibiting apoptosis via decreased binding to Apaf-1, caspase cleavage and activity, and cardiolipin peroxidase activity. These results are discussed with X-ray crystallography structures of K39 acetylated (1.50 Å) and acetylmimetic K39Q Cyt*c* (1.36 Å) and NMR dynamics. We propose that K39 acetylation is an adaptive response that controls electron transport chain flux, allowing skeletal muscle to meet heightened energy demand while simultaneously providing the tissue with robust resilience to ischemia-reperfusion injury.

Cytochrome *c* (Cyt*c*) is a small, 104 amino acid, essential protein with a covalently linked heme group that performs vital functions in both life-sustaining and cell death pathways. In the mitochondrial inter-membrane space (IMS), it acts as a single electron carrier from complex III to complex IV, also known as cytochrome *c* oxidase (COX), as part of the electron transport chain (ETC)[1]. Apart from respiration, Cyt*c* plays a key role in intrinsic apoptosis, where its release from the IMS into the cytosol allows Cyt*c* to bind apoptosis protease activating factor-1 (Apaf-1), activating the apoptosome, which in turn activates caspase-9 and the downstream caspase cascade[2-4]. Cyt*c* possesses

[1]Center for Molecular Medicine and Genetics, Wayne State University, Detroit, MI 48201, USA. [2]Instituto de Investigaciones Químicas, Universidad de Sevilla - CSIC, 41092 Sevilla, Spain. [3]Department of Biochemistry, Microbiology, and Immunology, Wayne State University, Detroit, MI 48201, USA. [4]College of Medicine, Dankook University, Cheonan-si Chungcheongnam-do 31116, Republic of Korea. [5]Life Sciences Collaborative Access Team, Northwestern University, Center for Synchrotron Research, Argonne, IL 60439, USA. [6]Department of Emergency Medicine, University of Michigan Medical School, Ann Arbor, MI 48109, USA. [7]Department of Health Care Sciences, Eugene Applebaum College of Pharmacy & Health Sciences, Wayne State University, Detroit, MI 48201, USA. [8]Max Planck Institute for Molecular Genetics, 14195 Berlin, Germany. [9]These authors contributed equally: Paul T. Morse, Gonzalo Pérez-Mejías. ✉e-mail: idiazmoreno@us.es; mhuttema@med.wayne.edu

cardiolipin peroxidase activity, another function related to cell death[5]. Cardiolipin is a lipid that makes up about 20% of the inner mitochondrial membrane (IMM) lipid composition. During apoptosis, cardiolipin is oxidized by Cyt*c*, which assists in outer membrane permeabilization[6,7]. Additionally, Cyt*c* scavenges reactive oxygen species (ROS)[8,9]. Given the numerous critical processes carried out by Cyt*c*, the protein is tightly regulated by several mechanisms: allosteric regulation by ATP, expression of tissue-specific isoforms (somatic and testes), and post-translational modifications[10].

Six tissue-specific phosphorylation sites and one prostate cancer-specific acetylation site have been characterized on Cyt*c*. Five of the modifications have previously been found to reduce the capacity of Cyt*c* to perform its roles in respiration and apoptosis. The phosphorylations are tyrosine 97 in the heart[11,12], tyrosine 48 in the liver[13–15], threonine 28 and threonine 58 in the kidney[16–18], and serine 47 in the brain[18–20]. These phosphorylations were generally found under basal conditions and lost during ischemia, providing a mechanism for reperfusion injury when oxygen is reintroduced into the tissue. Another phosphorylation of Cyt*c* on threonine 49 (numbering based on mature Cyt*c* which lacks the start methionine) was reported to increase in aged mouse hearts[21], but, as the authors do not report knocking out endogenous Cyt*c* prior to transfecting in recombinant, phosphomimetic Cyt*c*, data interpretation regarding functional effects of this modification are limited. In prostate cancer, acetylation of lysine 53 promotes two central hallmarks of cancer: switching to Warburg metabolism and evasion of apoptosis[22,23].

Here, we describe the identification and characterization of an acetylation site on lysine 39 (K39) of skeletal muscle Cyt*c* purified from porcine tibialis anterior (TA) skeletal muscle. Uniquely, K39 acetylation was found after ischemia and was absent under basal conditions. Much like other tissue types, skeletal muscle can experience ischemia-reperfusion injury. Total knee arthroplasty and other conditions that require the application of a tourniquet commonly cause ischemia-reperfusion injury in skeletal muscle[24,25]. Other conditions, such as peripheral artery disease or compartment syndrome, also induce ischemic changes in skeletal muscle[26,27]. Despite this, skeletal muscle is uniquely resistant to ischemia-reperfusion injury compared to other tissues, such as the brain[28,29]. Using purified acetylated and recombinant acetylmimetic protein and a mammalian cell overexpression acetylmimetic system, we studied the effects of this modification on Cyt*c* function. We report that this acetylation stimulates respiration in the ischemic muscle to meet the increased energy demand. Additionally, this modification reduces the pro-apoptotic capabilities of the protein, protecting the skeletal muscle under conditions of stress. In conjunction with X-ray crystallography and nuclear magnetic resonance (NMR) studies of our Cyt*c* variants, we propose that K39 acetylation of Cyt*c* contributes to the resilience of skeletal muscle to ischemia-reperfusion injury.

## Results

### Tibialis anterior muscle Cyt*c* is acetylated on lysine 39 after ischemia

Porcine tibialis anterior (TA) muscle was harvested and either immediately snap-frozen (control TA) or exposed to ischemic treatment prior to being snap-frozen (ischemic TA). Cyt*c* was purified from 6 control and 6 ischemic TA samples (3 male and 3 female *Sus domesticus* per group; 6 months old) under conditions that preserve post-translational modifications. Mass spectrometry analysis revealed an acetylation site on K39 of Cyt*c* that was not present in any of the control samples but was detected in 3 of the ischemic TA samples (1 male and 2 female), making this an ischemia-specific acetylation (Fig. 1A). All mass spectrometry data have been deposited to the ProteomeXchange Consortium (http://proteomecentral.proteomexchange.org) via the PRIDE[30] partner repository, with the dataset identifier PXD040915. Note: In the raw mass spectrometry results, amino acids were numbered

including the start methionine (mature Cyt*c* lacks the start methionine), therefore the numbering will be one number higher than what is reported here for the mature protein. Lysine acetylation of Cyt*c* was also identified on K27, K79, and K86 in some samples, however, these acetylations were present in both the control and ischemic TA samples (Supplementary Table 2). K88 acetylation was also identified, but the intensity was so low that occupancy could not be calculated.

K39 acetylation was never present in the control TA samples and was gained during ischemia, making it an adaptive response to ischemia. K39 acetylation was not present when control TA muscle Cyt*c* was analyzed by mass spectrometry or western blot (Fig. 1B). Acetylation of TA muscle Cyt*c* was induced at 45 min of ischemia and appeared to increase slightly at 60 min of ischemia (Fig. 1C). The acetylation of Cyt*c* purified from ischemic TA muscle was removed in vitro by sirtuin5 (Fig. 1D), a mitochondria-localized deacetylase.

### Bacterial overexpression and protein purification of WT, acetylmimetic K39Q, and additional Cyt*c* variants

To study the effects of K39 acetylation in vitro, a recombinant purified protein system was employed as we have previously done for lysine 53 acetylation in prostate cancer[22,23]. The overexpressed wild-type protein (WT) is nonacetylated. To mimic lysine acetylation, K39 was mutated to a glutamine residue (K39Q), which is a polar, uncharged residue like acetyl-lysine. Using glutamine as a mimetic for acetyl-lysine is an established technique for both Cyt*c*[22,23] and other proteins in general due to their similar side chains, which are both uncharged amides[31–34]. As another control, K39 was replaced with arginine (K39R), which retains a positive charge at the residue like unacetylated lysine. As an additional experimental control, K39 was mutated to a glutamate residue (K39E), which places a negative charge at the residue. Recombinant WT, K39R, K39Q, and K39E proteins were purified after bacterial overexpression. LabSafe GEL Blue staining of the recombinant proteins resolved on an SDS polyacrylamide gel demonstrates high purity of Cyt*c* (Fig. 1E). The UV-Vis spectra of the reduced recombinant proteins also demonstrate purity and proper folding due to the presence of the characteristic α, β, and γ peaks (Fig. 1F; note: the reduced K39Q spectrum overlaps with the reduced K39R spectrum). The presence of the 695 nm peak in the oxidized spectra indicates successful incorporation and correct coordination of the heme iron in all mutants (Fig. 1F, insert).

### Acetylmimetic K39Q Cyt*c* and in vivo acetylated porcine TA muscle Cyt*c* result in increased cytochrome *c* oxidase activity and decreased caspase-3 activity

Regulatory-competent COX was purified from porcine heart under conditions that preserve post-translational modifications, and COX activity was measured with a Clark-type oxygen electrode. The maximal oxygen consumption rates using recombinant K39Q and K39E proteins were increased by 38% and 72%, respectively, compared to recombinant WT protein (Fig. 1G). Similar measurements with in vivo K39 acetylated Cyt*c* isolated from ischemic porcine TA muscle demonstrated an oxygen consumption rate increase of 58% compared to in vivo unacetylated Cyt*c* isolated from control porcine TA muscle (Fig. 1H). The interaction of Cyt*c* and COX is primarily electrostatic in nature[35,36]. Thus, to test the hypothesis that change in charge at residue K39 drives maximal functional effects, the K39E mutant was included. Given that lysine is positively charged, glutamine is neutral but polar like acetyl-lysine, and glutamate is negatively charged, the K39Q and K39E replacements produce a change of 1 and 2 charge units compared to WT, respectively. As predicted, K39E Cyt*c* shows a more pronounced effect to activate COX activity compared to K39Q and WT, producing the trend: WT < K39Q < K39E Cyt*c* (Fig. 1G). Overall, these data indicate that acetylation increases oxygen consumption rate and that K39Q replacement is a good mimetic for acetylated K39.

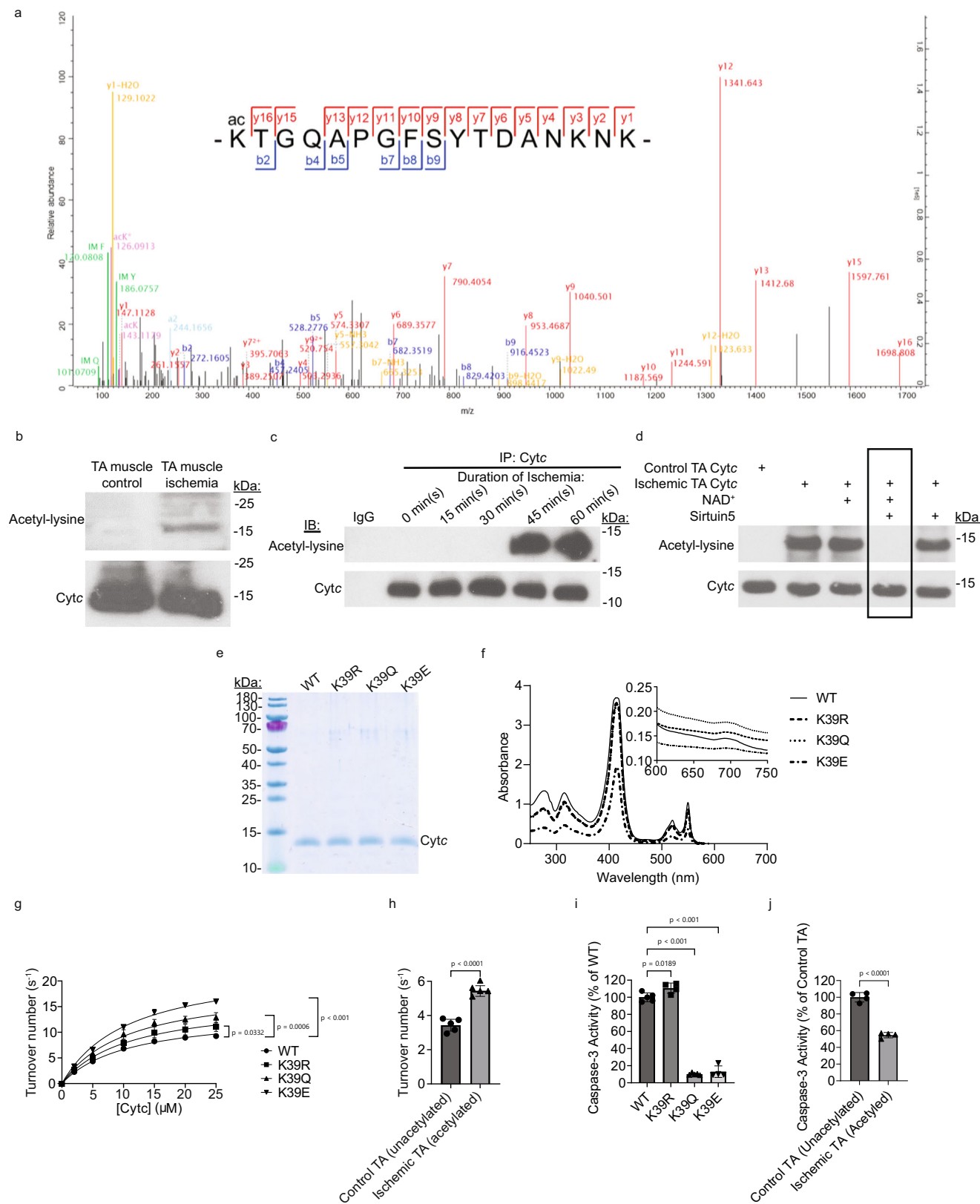

Previous research on the structure of the apoptosome has indicated that K39 is a part of the Apaf-1 binding domain[3]. Therefore, K39Q acetylmimetic and in vivo K39 acetylated Cyt*c* could regulate apoptosis. Interestingly, the caspase-3 activities of recombinant K39Q and K39E proteins demonstrated a significant decrease of 90% and 87%, respectively, compared to recombinant WT protein (Fig. 1I), indicating

that K39 acetylation protects the cells from cell death. The caspase-3 activity of the in vivo K39 acetylated Cyt*c* isolated from ischemic porcine TA muscle decreased caspase-3 activity by 45% compared to the in vivo unacetylated Cyt*c* isolated from control porcine TA muscle (Fig. 1J). These data confirm that K39Q is a good mimetic for acetylated K39 and that K39 serves as an important regulator for binding Apaf-1.

**Fig. 1 | Ischemic TA muscle Cyt*c* is acetylated on lysine 39 and acetylmimetic K39Q and acetylated Cyt*c* increases COX activity and decreases caspase-3 activity. A** Representative mass spectrum of Cyt*c* peptide K(acetyl)TGQAPGF-SYTDANKNK identifying Cyt*c* K39 acetylation in ischemic, but not control, samples (n = 12: 6 control TA (3 male and 3 female) and 6 ischemic TA (3 male and 3 female with 1 male and 2 female samples demonstrating K39 acetylation); Proteo-meXchange Consortium, PRIDE: PXD040915). **B** Cyt*c* is acetylated in ischemic but not control TA muscle as shown after immunoprobing for acetyl-lysine and Cyt*c* (n = 3). **C** Representative IP-Immunoblot experiment of porcine TA muscle showing acetylation state of Cyt*c* at 0, 15, 30, 45, and 60 min of ischemia (n = 3). **D** Representative in vitro deacetylation assay showing that sirtuin5 removes acetylation from Cyt*c* (n = 3). **E** Representative LabSafe GEL Blue stained 10% tris-tricine SDS-PAGE gel showing purity of the recombinant Cyt*c* purified from bacteria after overexpression (n = 3). **F** Reduced Cyt*c* spectra and oxidized Cyt*c* spectra (inset) of recombinant Cyt*c* indicate correct folding of the proteins (the reduced K39Q spectrum overlaps with the reduced K39R spectrum). Oxygen consumption rate of 26.7 nM pig heart COX was measured using an Oxygraph+ system at 25 °C. **G** The recombinant WT, K39R, K39Q, and K39E proteins (n = 3) or **H** control and ischemic purified pig TA muscle Cyt*c* samples (n = 5) were titrated at concentrations of 0, 2, 5, 10, 15, 20, and 25 μM or injected at 5 μM, respectively. For G, a one-way ANOVA comparing the mean of each mutant with the mean of the control mutant (WT) at the 25 μM condition with the Dunnett post-hoc test was used. For H, a student's two-tailed t-test assuming equal variance was used. **I** Cytosolic extracts from Cyt*c* knockout embryonic fibroblasts were incubated with the recombinant WT, K39R, K39Q, and K39E proteins (n = 5 for WT, K39Q; n = 4 for K39R, K39E) or **(J)** control and ischemic purified pig TA muscle Cyt*c* samples (n = 4) for 2.5 h at 37 °C. Rho-damine fluorescence that resulted from caspase-3 mediated cleavage of Z-DEVD-R110 was used as a measure of caspase-3 activity. For I, a one-way ANOVA comparing the mean of each mutant with the mean of the control mutant (WT) with the Dunnett post-hoc test was used. For J, a student's two-tailed t-test assuming equal variance was used. Data are represented as means ± standard deviation.

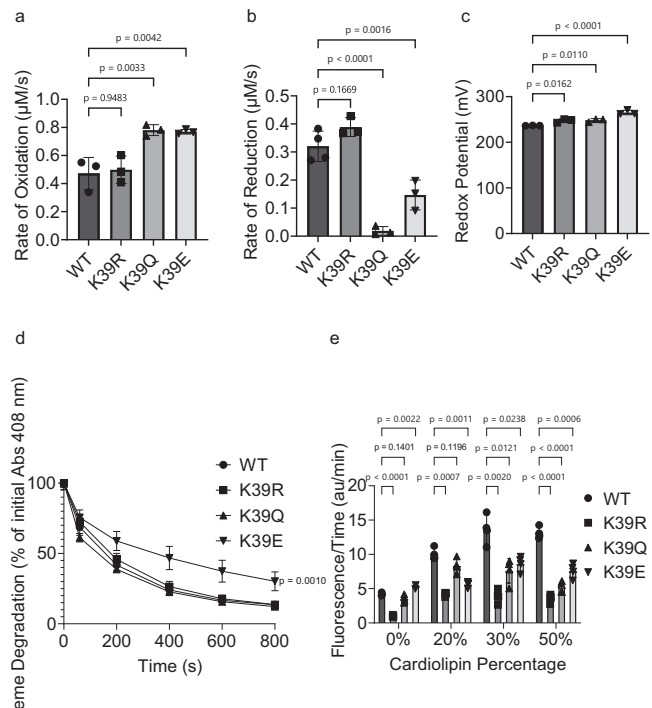

**Fig. 2 | Acetylmimetic K39Q Cyt*c* demonstrates altered rate of oxidation by H₂O₂, rate of reduction by superoxide, midpoint redox potential, and cardiolipin peroxidase activity. A** Initial rate of oxidation of reduced recombinant Cyt*c* proteins by 100 μM H₂O₂ (n = 3). **B** Initial rate of reduction of oxidized recombinant Cyt*c* proteins by superoxide generated with a hypoxanthine/xanthine oxidase system (n = 4). **C** Midpoint redox potentials of recombinant Cyt*c* proteins were measured using the equilibration method with DCIP as the reference compound (n = 3). **D** Heme degradations of oxidized recombinant Cyt*c* proteins after the addition of 3 mM H₂O₂ (n = 3). **E** Cardiolipin peroxidase activities of recombinant Cyt*c* proteins with liposomes containing 0%, 20%, 30%, and 50% tetrazolyl-cardiolipin (TOCL) were measured using resorufin fluorescence after the addition of 5 μM H₂O₂ (n = 4). Data are represented as means ± standard deviation. A one-way ANOVA comparing the mean of each mutant with the mean of the control mutant (WT) with the Dunnett post-hoc test was used. For D, the 800 s condition specifically was compared.

## Acetylmimetic K39Q Cyt*c* demonstrates an increased rate of oxidation and decreased rate of reduction

In addition to respiration and apoptosis, Cyt*c* also functions as a ROS scavenger. Two major types of ROS that Cyt*c* detoxifies in the IMS are superoxide and H₂O₂. The effect of K39 acetylation on ROS scavenging was studied in reaction with H₂O₂ (rate of oxidation) and superoxide (rate of reduction). To determine the rate of oxidation, recombinant ferro-(Fe²⁺)-Cyt*c* proteins were reacted with 100 μM H₂O₂, and the initial reaction was monitored spectrophotometrically via the absorbance of the α peak at 550 nm which is present in reduced, ferro-(Fe²⁺)-Cyt*c* and greatly diminished in the oxidized, ferri-(Fe³⁺)-Cyt*c*. The rates of oxidation of K39Q and K39E increased by 66% and 63%, respectively, compared to WT (Fig. 2A). This indicates that K39Q Cyt*c* is a better scavenger of H₂O₂ than WT.

Superoxide is a physiologically relevant source of mitochondrial ROS that is produced at complexes I and III[37]. For the rate of reduction, superoxide was generated by the hypoxanthine/xanthine oxidase reaction system. To determine the rate of reduction, recombinant ferri-(Fe³⁺)-Cyt*c* proteins were reacted with superoxide, and the initial reaction was monitored spectrophotometrically at 550 nm. The rates of reduction of recombinant K39Q and K39E Cyt*c* decreased 94% and 54%, respectively, compared to WT (Fig. 2B).

## Acetylmimetic K39Q Cyt*c* demonstrates increased redox potential

The midpoint redox potential of native Cyt*c* is between that of complex III and COX, facilitating efficient electron transfer. The literature range for native Cyt*c* midpoint redox potentials is 220 to 270 mV[38]. The midpoint redox potentials for the recombinant WT, K39R, K39Q, and K39E proteins were measured spectrophotometrically using the equilibration method[39] and the measured values fell within the reported range (Fig. 2C).

## Acetylmimetic K39Q Cyt*c* demonstrates no change in heme stability

High levels of ROS cause Cyt*c* to lose functionality by degrading the catalytic heme moiety. The stabilities of the heme group were measured spectrophotometrically by tracking the dissipation of the characteristic heme Soret peak at 408 nm after the challenge with 3 mM H₂O₂. There was no difference in heme degradation between recombinant WT, K39R, and K39Q proteins, which decreased 86%, 86%, and 88%, respectively, in the Soret peak absorbance at 800 s (Fig. 2D). However, recombinant K39E protein demonstrated a significant reduction in heme degradation, with only a 70% decrease in the Soret peak absorbance at 800 s.

## Acetylmimetic K39Q Cyt*c* demonstrates reduced cardiolipin peroxidase activity

A secondary pro-apoptotic function of Cyt*c* is its cardiolipin peroxidase activity. During apoptosis, the peroxidase activity of Cyt*c* is increased, and cardiolipin peroxidation can occur in the presence of H₂O₂ catalyzed by Cyt*c*. This process facilitates the release of Cyt*c* into the cytosol, committing the cell to apoptosis[40]. The cardiolipin peroxidase activities were measured via the fluorescence of resorufin. The

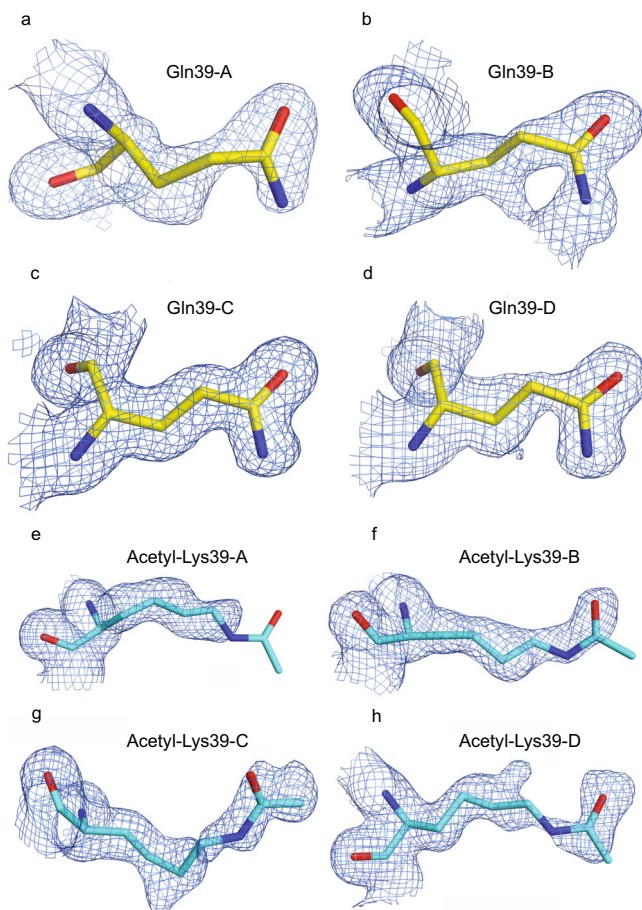

**Fig. 3 | Omit density maps of K39Q and K39 acetylated Cyt*c* crystal structures at residue 39. A**–**D** Omit density maps of chain A through D at residue 39 of K39Q crystal structure (8DZL.pdb, 1.36 Å). **E**–**H** Omit density maps of chain A through D at residue 39 of ischemic porcine Cyt*c* crystal structure (8DVX.pdb, 1.50 Å). The refined, fractional occupancies for the four acetyl groups are 0.37, 0.80, 0.95, and 0.89, respectively for chain-A through chain-D. All omit density maps are contoured at 0.5 RMSD.

Cyt*c* proteins oxidize cardiolipin, which in turn oxidizes Amplex red to resorufin. The cardiolipin peroxidase activity of recombinant K39Q Cyt*c* was significantly decreased compared to recombinant WT protein, ranging from 17% to 60% decreased fluorescent signal in lipid vesicles composed of 20%, 30%, and 50% cardiolipin (Fig. 2E). These data, along with the reduction in caspase-3 activity reported above for acetylated and acetylmimetic Cyt*c*, support an anti-apoptotic role of K39 acetylation.

## Crystallography results

Acetylmimetic K39Q and K39 acetylated Cyt*c* proteins were crystallized and analyzed via X-ray crystallography. The K39Q structure (Supplementary Table 1: 8DZL) at 1.36 Å has continuous omit map density for residue 39 in chain-C when contoured at 1.0 root mean square deviation (RMSD) and for all four chains (chains A through D) when contoured at 0.7 RMSD (Fig. 3A–3D). The respective backbone RMSD values for the four molecules (chains A through D) relative to chain A of the native structure (5C0Z.pdb) are 0.279, 0.323, 0.362, and 0.337. The overall real space correlation coefficient (RSCC) is 0.928. The only disordered residues are Lys25-His26 in chains C and D which have broken density for their backbone atoms at 1.0 RMSD and RSCC values ranging from 0.624 to 0.767. Their objectively refined conformations match the conformations for chains A and B in the K39Q structure and chain A in the native protein. The

structure of porcine K39 acetylated Cyt*c* (Supplementary Table 1: 8VDX) at 1.5 Å has a lower overall RSCC value of 0.890, but all residues have continuous backbone density at 1.0 RMSD. The fractional occupancies for the four acetyl groups (Fig. 3E–3H) have an average value of 0.75 ± 0.26 as shown by Phenix.Refine script (Supplementary Fig. 1A). The respective backbone RMSD values for the four molecules (chains A through D) relative to chain A of the native structure (5C0Z.pdb) are 0.260, 0.307, 0.274, and 0.266. As anticipated, given the longer side chain, the K39 acetylated Cyt*c* demonstrates greater variation at residue 39 compared to the K39Q Cyt*c* (Supplementary Figure 1B, 1C). Interestingly, acetylmimetic mutations, K39Q (8DZL) and K53Q (7LJX)[22], tend to more globally perturb the entire Cyt*c* structure compared to phosphomimetic mutations, T28E (5DF5)[16] and S47E (6N1O)[19], which show much fewer differences compared to native, rodent WT (5C0Z)[19] (Supplementary Figure 1D). Specifically, both the K39Q and K53Q structures have variation in the D50-to-G60 and N70-to-E90 regions, while the T28E and S47E structures have greatly reduced variability compared to the WT in these regions and globally, highlighting that acetylation may have stronger effects on Cyt*c* structure and function than phosphorylation.

For NMR analyses and electrostatic surface potentials, there was a switch-like effect based whether K39 was positively charged or not with recombinant WT and K39R Cyt*c* behaving in one manner while K39Q and K39E Cyt*c* behaved in a different manner. (Supplementary Fig. 2). Prior to measurement, protein purity was confirmed via Coomassie blue staining (Supplementary Fig. 3A). Proper protein folding and heme coordination were evaluated by 1D ¹H NMR spectra (Supplementary Fig. 3B, C). Point mutations and monomerization state were confirmed via tryptic digestion analyses and dynamic light scattering, respectively (Supplementary Fig. 4). The chemical shift perturbations (CSP) of the amide backbone signals of the 2D ¹⁵N-¹H Heteronuclear Single Quantum Correlation (HSQC) NMR spectra between the recombinant WT protein and the recombinant K39R, K39Q, and K39E proteins were calculated. Resonance assignment for each recombinant protein was confirmed via recording 3D ¹⁵N-¹H Nuclear Overhauser Effect Spectroscopy (NOESY)-HSQC and 3D ¹⁵N-¹H TOtal Correlation SpectroscopY (TOCSY)-HSQC spectra (Supplementary Figs. 5–7). The magnitude of the perturbation from the WT spectra induced by the mutations followed the change in charge at the residue, as K39E demonstrated greater perturbation than K39Q which demonstrated greater perturbation than K39R. Interestingly, the mutations at residue 39 not only affected the chemical environment of nearby residues, but also residues located in the M80-containing loop and the K55-to-W59 stretch (Fig. 4A, B).

Mutation at K39 did not significantly alter the rotational correlational time ($\tau_c$) values (WT 5.75 ± 0.03 ns, K53Q 5.66 ± 0.02 ns), though these values are lower than that of the human WT Cyt*c* protein (6.33 ± 0.02 ns)[14]. However, the K39Q substitution slightly affected both relaxation rates ($R_1$ and $R_2$) parameters (Figs. 5A, B, Supplementary Figs. 8, 9). Comparing the differences in $R_1$, $R_2$, and ¹⁵N{¹H} NOE parameters revealed that the M80-containing loop of WT exhibited higher mobility in the ps-to-ns timescale, as the ¹⁵N{¹H} NOE values for the P76-to-K88 segment are smaller for the recombinant WT protein. Similar behavior is observed for the N-terminal and C-terminal α helices and the position of the mutation. Surprisingly, the K55-to-W59 segment, spatially close to the propionate HP7 of the heme group and to the proximity of the mutation, showed almost no difference at the ¹⁵N{¹H} NOE values for the recombinant acetylmimetic K39Q protein. However, in this region, the spin-lattice ($R_1$) and spin-spin ($R_2$) relaxation rates of the recombinant WT protein revealed a possible conformational exchange in the μs-to-ms timescale. Additional ¹⁵N{¹H} NOE experiments were recorded with the recombinant K39E and K39R proteins at 500 MHz (Supplementary Fig. 10).

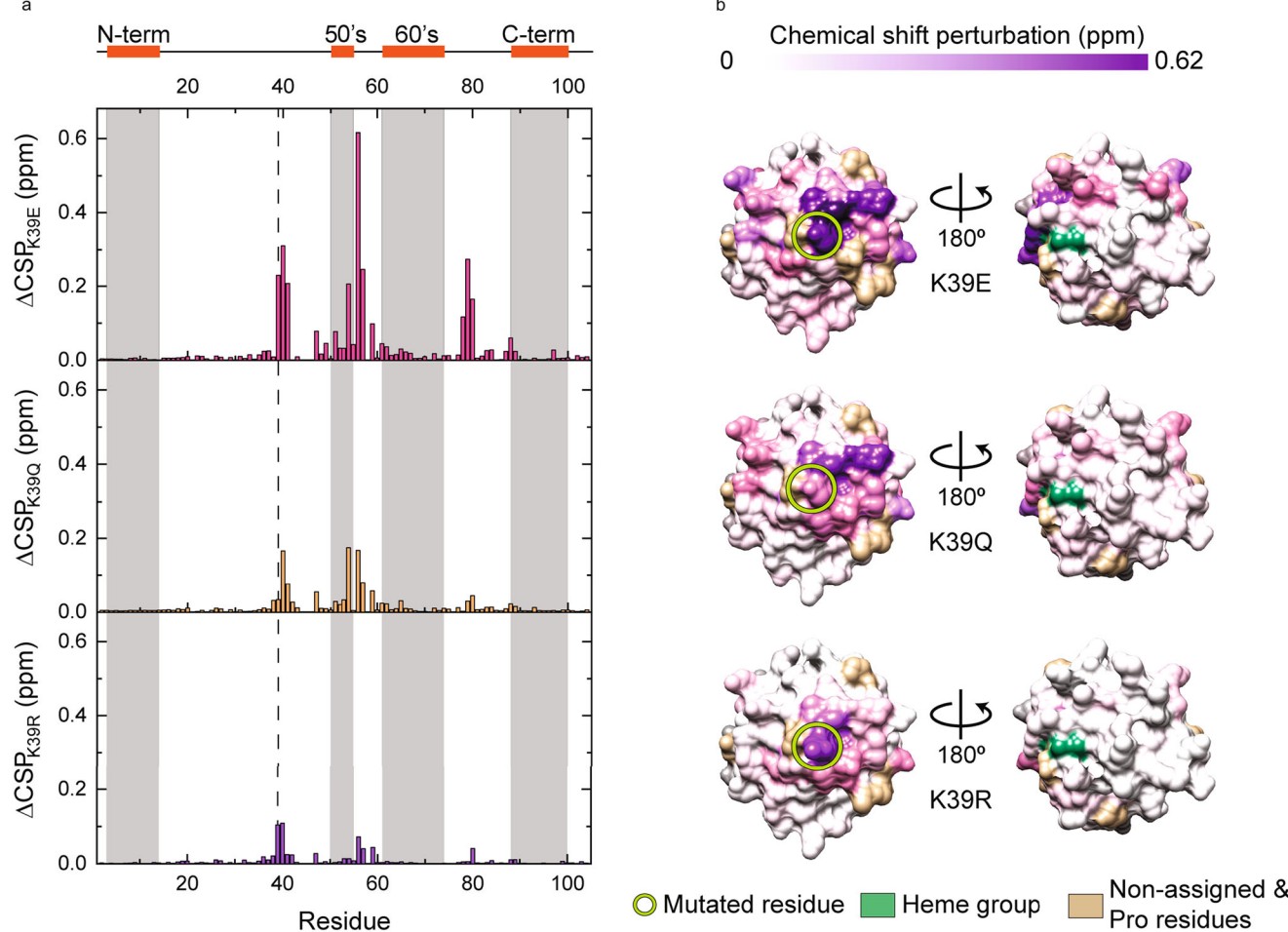

**Fig. 4 | Nuclear magnetic resonance structure analysis of Cyt*c* mutants.**
**A** Chemical shift perturbation (CSP) bar plot of Cyt*c* backbone amide signals ($\Delta CSP_X = CSP_X - CSP_{WT}$) of reduced Cyt*c* mutants (K39E in pink, K39Q in yellow, and K39R in purple). A scheme of the secondary structure elements is included at the top and the mutated position is indicated with dashed lines. **B** Surface maps of NMR $\Delta CSP$ of reduced Cyt*c* mutants. Each map shows residues colored according to $\Delta CSP$ values calculated previously. The Heme group is in green and non-assigned and proline residues are in tan. Source data are provided as a Source Data file.

## Cells expressing acetylmimetic K39Q Cyt*c* demonstrate activation of mitochondrial respiration

To further study the effects of K39 acetylation, we generated cell lines stably expressing WT, K39R, K39Q, and K39E Cyt*c*. These cell lines were created by transfecting the Cyt*c* constructs into Cyt*c* double knockout mouse embryonic fibroblasts. This double knockout system, where both the somatic and testes isoforms of Cyt*c* were knocked out, was used to ensure that knockout of the somatic isoform alone would not induce expression of the testes isoform[41]. Additionally, a Cyt*c*-null empty vector (EV) cell line was employed as an additional control for all cellular experiments. Clones of each Cyt*c* variant were selected based on similar expression of Cyt*c* (Fig. 6A). Mitochondrial biogenesis was found to be similar between the cell lines as assessed via western blot for mitochondrial transcription factor A (TFAM), peroxisome proliferator-activated receptor gamma coactivator 1-alpha (PGC-1α), and citrate synthase (CS) (Supplementary Fig. 11A). The cell line expressing K39Q Cyt*c* demonstrated a 51% increase in ATP content compared to the WT (Fig. 6B). Intact cellular respiration was measured using a Seahorse bioanalyzer. The reaction between Cyt*c* and COX is generally considered to be the rate-limiting step of the electron transport chain (reviewed in ref. 10). Therefore, mutations to Cyt*c* may impact intact cellular respiration. The basal oxygen consumption rate of the K39Q cell line demonstrated a 70% increase compared to WT (Fig. 6C, D). Additionally, the K39Q cell line showed increases in non-mitochondrial oxygen consumption, proton leak, ATP-coupled respiration, maximal respiration rate, and spare respiratory capacity compared to WT (Supplementary Fig. 12). Altogether, these data indicate that K39Q acetylmimetic replacement enhances electron flow through the electron transport chain resulting in an increase in cellular respiration.

## Cells expressing acetylmimetic K39Q Cyt*c* have higher mitochondrial membrane potential which corresponds to higher ROS production

Given that the in vitro protein and intact cellular data indicated an increase in respiration due to K39 acetylation or acetylmimetic replacement, we hypothesized that this would translate to an increase in the mitochondrial membrane potential ($\Delta\Psi_m$). $\Delta\Psi_m$ levels of the cell lines were measured using JC-10, a ratiometric probe. Overall, the trend in $\Delta\Psi_m$ matched the trend in respiration with EV < WT < K39R < K39Q < K39E. $\Delta\Psi_m$ of the K39Q Cyt*c* cell line demonstrated a 42% increase in red/green fluorescence compared to WT, indicating higher $\Delta\Psi_m$ (Fig. 6E).

Previous research has established a relationship between $\Delta\Psi_m$ and mitochondrial ROS production. Mitochondrial ROS levels of the cell lines were measured using MitoSOX. The production of mitochondrial ROS matched the trend in respiration rate and $\Delta\Psi_m$ with increasing ROS levels for EV < WT < K39R < K39Q < K39E. MitoSOX fluorescence of the K39Q cell line increased 50% compared to WT (Fig. 6F). Altogether, these data suggest that K39Q acetylmimetic replacement promotes respiration, which in turn increases $\Delta\Psi_m$ and mitochondrial ROS production.

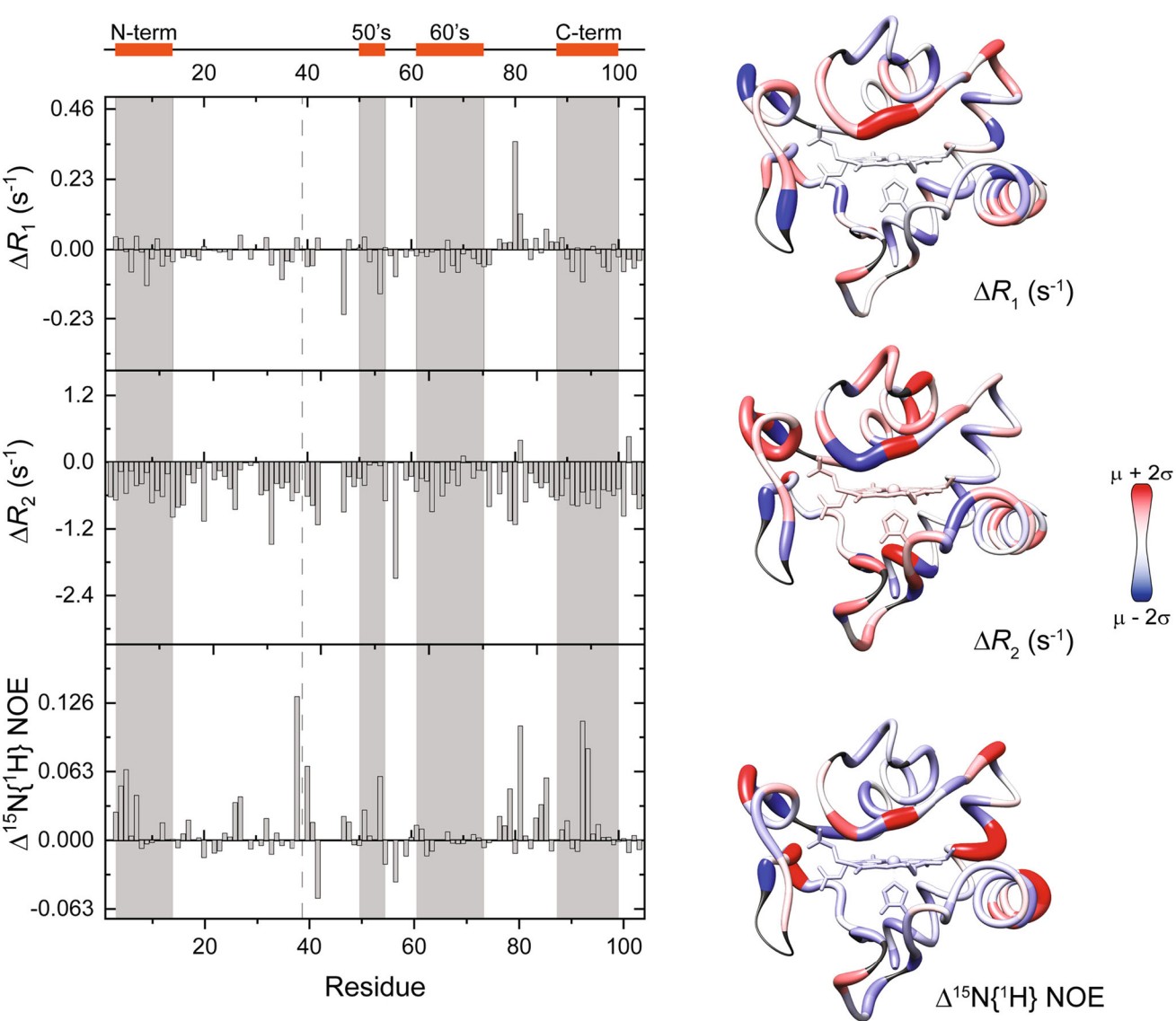

**Fig. 5 | Difference between WT and acetylmimetic K39Q nuclear magnetic resonance parameters. A** Average differences (means) in longitudinal relaxation rate $R_1$ (top), transversal relaxation rate $R_2$ (middle), and heteronuclear $^{15}N\{^1H\}$ NOE (bottom) between the experimental values at 500 MHz for the reduced forms of K39Q and WT Cyt*c*, plotted as a function of the residue number (n = 2). Each difference was calculated from a set of relaxation parameters measured on two biologically independent samples for WT and K39Q Cyt*c*. Raw data is represented in Supplementary Fig. 8. A scheme of the secondary structure elements is included at the top and the mutated position is indicated with dashed lines. **B** Ribbon structure of K39Q Cyt*c* colored according to the difference in its dynamic properties from blue (decrease) to red (increase) compared with WT protein. Undetectable or overlapping backbone resonances are in gray. Source data are provided as a Source Data file.

## Cells expressing acetylmimetic K39Q Cyt*c* demonstrate increased mitochondrial membrane potential but not ROS production after oxygen-glucose deprivation/reoxygenation

After exposure to stressful conditions, such as reperfusion following ischemia, $\Delta\Psi_m$ increases due to ETC activation, leading to an exponential increase in mitochondrial ROS[42]. Because K39 acetylation was present only after ischemia, but not under normal conditions, the role of this modification was further assessed using an oxygen-glucose deprivation/reoxygenation (OGD/R) model[43], which simulates ischemia-reperfusion injury in cell culture as previously described[20].

$\Delta\Psi_m$ and mitochondrial ROS levels of the cell lines were measured at normoxia (control) and after OGD/R. For $\Delta\Psi_m$, each cell line showed a significant change in JC-10 red/green fluorescence ratio after exposure to OGD/R compared to the normoxia control. Interestingly, following OGD/R, the K39Q cell line demonstrated an increase of only 4%

JC-10 red/green fluorescence compared to the WT cell line, mitigating the larger difference seen when the experiment is run under conditions of normoxia (Fig. 6G). For mitochondrial ROS, all cell lines except K39Q show a significant change in MitoSOX fluorescence after OGD/R exposure compared to normoxia control (Fig. 6H). These data show that K39Q Cyt*c* increases respiration, $\Delta\Psi_m$, and ROS already under baseline conditions versus WT, but it does not lead to a further increase following OGD/R, potentially protecting the cells from additional ROS upon reperfusion.

## Cells expressing acetylmimetic K39Q Cyt*c* are better protected from cell death after exposure to H₂O₂, prolonged OGD/R, or ER stress

Cell death of the cell lines was analyzed via annexin V/propidium iodide (PI) staining by flow cytometry following treatment with H₂O₂,

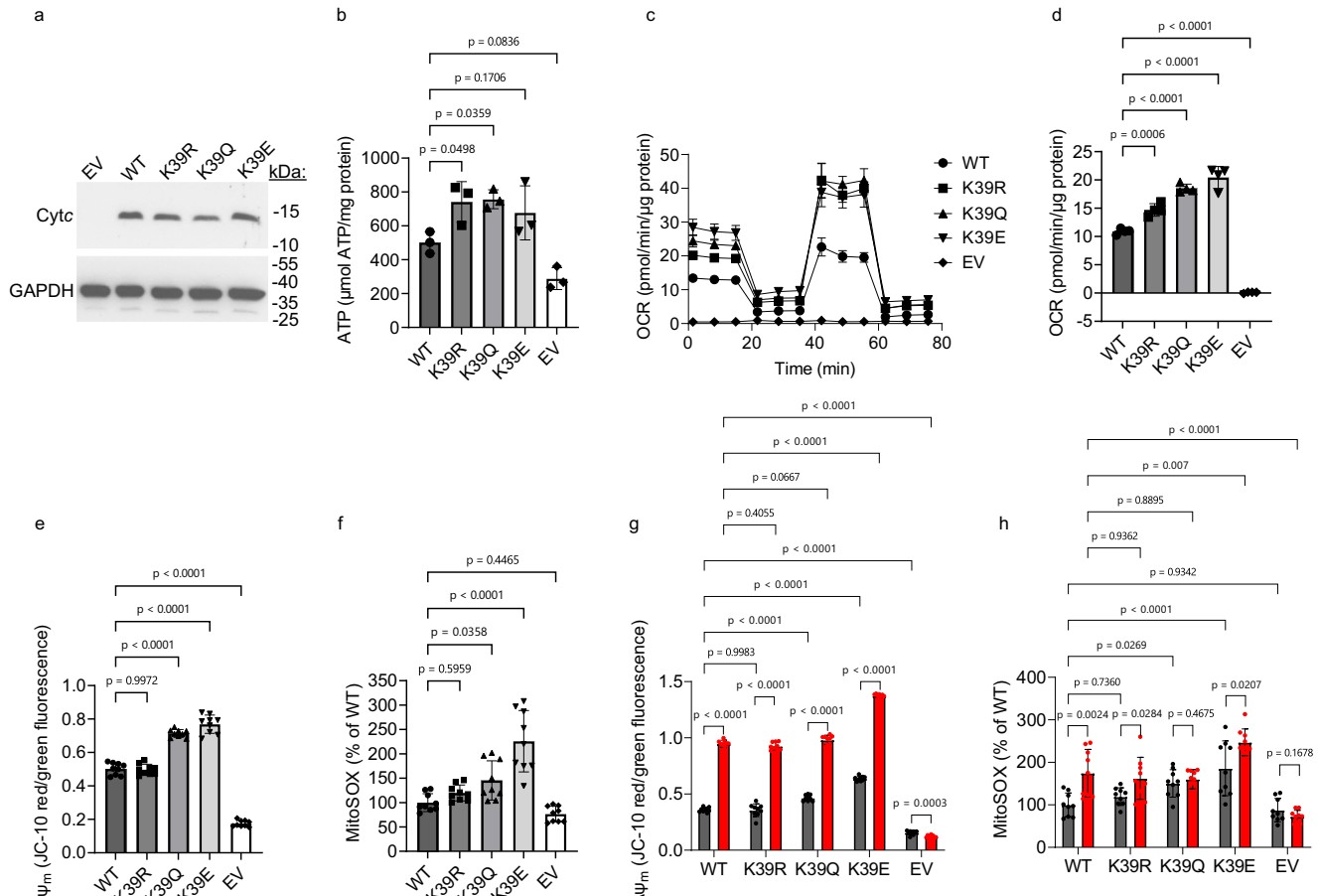

**Fig. 6 | Acetylmimetic K39Q Cyt*c* expressing cells show increased respiration, mitochondrial membrane potential, and ROS production under basal conditions and increased mitochondrial membrane potential but not ROS production after oxygen-glucose deprivation/reoxygenation. A** Representative western blot of Cyt*c* double knockout cells stably transfected with EV, WT, K39R, K39Q, and K39E constructs immunoprobed for Cyt*c* and loading control GAPDH (n = 3). **B** ATP levels in the stable cell lines (n = 3). **C** Mitochondrial stress test measured in Seahorse media supplemented with 10 mM glucose and 10 mM pyruvate was performed via sequential injections of 1 μM oligomycin, 2.5 μM carbonylcyanide-3-chlorophenylhydrazone (CCCP), and 1 μM rotenone/antimycin A (n = 4). **D** Basal oxygen consumption rate (OCR) from mitochondrial stress test (n = 4). **E** Mitochondrial membrane potentials (ΔΨ$_m$) of cells stably expressing WT, K39R,

K39Q, K39E Cyt*c*, and EV were measured using the red/green fluorescence ratio of the JC-10 probe (n = 9). **F** Mitochondrial ROS production of cells stably expressing WT, K39R, K39Q, K39E Cyt*c*, and EV were measured using MitoSOX fluorescence and normalized to total protein content (n = 9). **G**, **H** ΔΨ$_m$ and mitochondrial ROS production were measured as above at normoxia (gray bars) or after 90 min of oxygen-glucose deprivation followed by 30 min of reoxygenation (red bars) (n = 9). Data are represented as means ± standard deviation. A one-way ANOVA comparing the mean of each mutant with the mean of the control mutant (WT) with the Dunnett post-hoc test was used. For G and H, a student's two-tailed t-test assuming equal variance was used to compare the normoxia vs. OGD/R condition within each mutant.

OGD/R, or thapsigargin[44]. Treatment with 400 μM H$_2$O$_2$ for 16 h resulted in significantly lower total cell death of the K39Q cell line with 29% cell death compared to WT with 38% cell death (Fig. 7A). Treatment with oxygen-glucose deprivation for 16 h followed by 1 h of reoxygenation resulted in significantly lower total cell death of the K39Q cell line with 22% cell death compared to WT with 29% cell death (Fig. 7B). The highly glycolytic EV cell line appears unable to survive prolonged glucose deprivation, with an increased level of total cell death at 64%. Additionally, treatment with 1 mM thapsigargin for 24 h, which causes ER stress, also resulted in significantly lower total cell death of the K39Q cell line with 33% cell death compared to WT with 49% cell death (Fig. 7C). The representative flow cytometry gating strategy is available in the supplement (Supplementary Fig. 13). Overall, the decrease in total cell death for the K39Q cell line after each of the three treatments is similar to that seen with the in vitro caspase-3 activity using the recombinant K39Q Cyt*c* protein and K39 acetylated ischemic porcine TA muscle Cyt*c*. Altogether, this highlights the role of K39 acetylation in reducing the apoptotic capabilities of Cyt*c*.

## Cells expressing acetylmimetic K39Q Cyt*c* show disrupted pro-apoptotic signaling

The binding of positively charged Cyt*c* to its negatively charged binding pocket on Apaf-1 is mediated largely by electrostatic interactions[3]. As discussed above, K39 of Cyt*c* is known to be part of the binding site to Apaf-1[45]. Removing the positive charge at K39 perturbs this interaction, which in turn reduces downstream apoptotic signaling. In line with this, the cell line expressing acetylmimetic K39Q Cyt*c* showed reduced levels of Cyt*c* pulled down after Apaf-1 immunoprecipitation compared to the cell line expressing the WT (Fig. 7D). This also translated into reduced cleavage of caspase-9 (Fig. 7E) and caspase-3 (Fig. 7F) in the cell line expressing the acetylmimetic K39Q compared to the WT after treatment with 8 h of staurosporine, a potent inducer of intrinsic apoptosis[17,46,47].

## Discussion

In this study, we found that porcine tibialis anterior muscle Cyt*c* is acetylated on K39 after 45 min of ischemia. No other types of post-translational modifications of Cyt*c* were identified beyond lysine

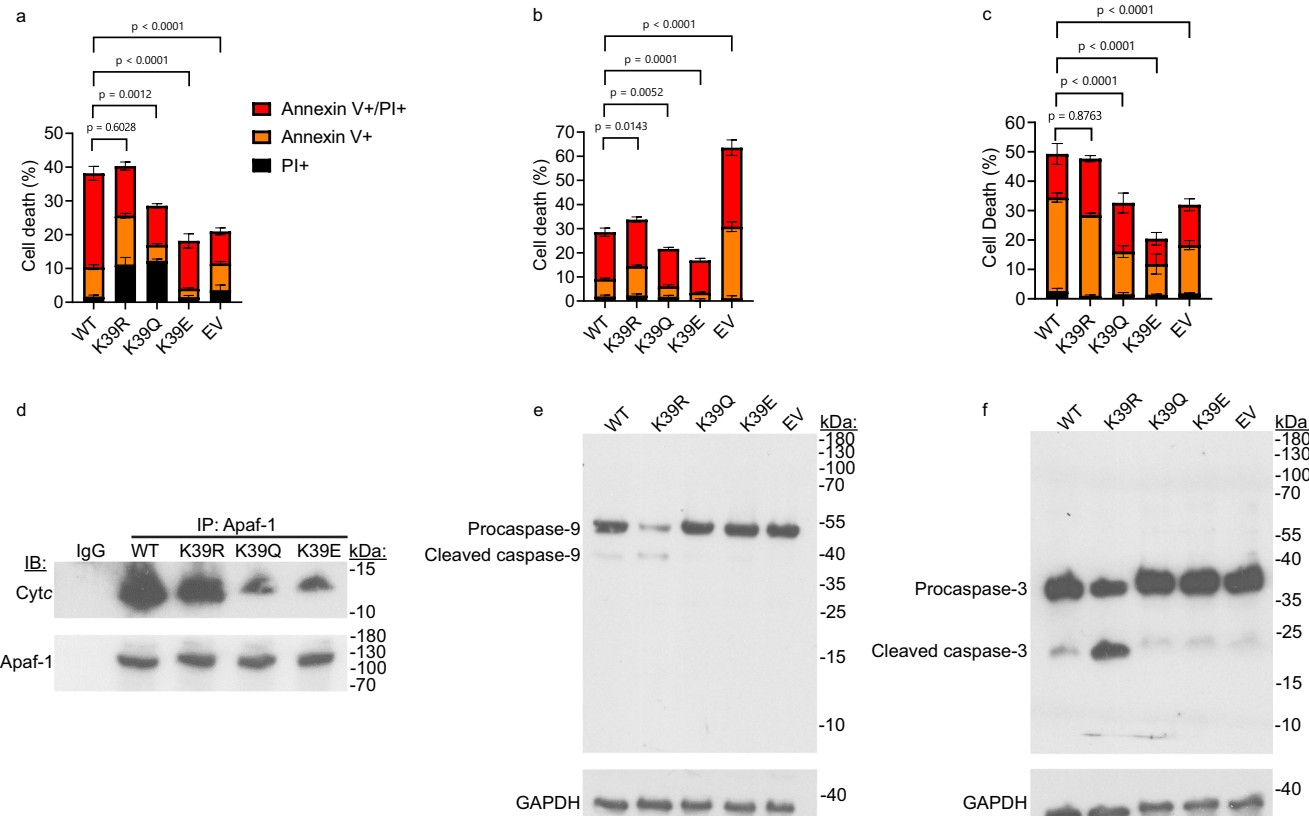

**Fig. 7 | Acetylmimetic K39Q Cytc expressing cells undergo less cell death and have defective apoptotic signaling. A** Annexin V/propidium iodide flow cytometry data for cells stably expressing WT, K39R, K39Q, K39E Cytc, and EV after exposure to 400 μM $H_2O_2$ for 16 h (n = 3), **B** 16 h of oxygen-glucose deprivation followed by 1 h of reoxygenation (n = 3), **C** or 1 mM thapsigargin for 24 h (n = 3). **D** Representative co-IP-Immunoblot experiment of Cytc double knockout cells stably transfected with WT, K39R, K39Q, and K39E constructs immunoprobed for Cytc and loading control Apaf-1 (n = 3). **E** Representative western blot of Cytc double knockout cells stably transfected with EV, WT, K39R, K39Q, and K39E constructs immunoprobed for caspase-9 and loading control GAPDH after exposure to 1 μM staurosporine for 8 h (n = 3). **F** Representative western blot of Cytc double knockout cells stably transfected with EV, WT, K39R, K39Q, and K39E constructs immunoprobed for caspase-3 and loading control GAPDH after exposure to 1 μM staurosporine for 8 h (n = 3). Data are represented as means ± standard deviation. A one-way ANOVA comparing the total cell death mean of each mutant with the total cell death mean of the control mutant (WT) with the Dunnett post-hoc test was used.

acetylation. The identification of this acetylation, which is gained during ischemia, was unexpected because our previous work in other tissues revealed Cytc phosphorylation under normal conditions, which was lost during ischemia. We have shown that phosphorylation is an important regulatory modification on Cytc present under physiological conditions which inhibits respiration and/or apoptosis. However, because these phosphorylations are lost during ischemia, we proposed that their loss causes reperfusion injury through increased ETC activity, $\Delta\Psi_m$ hyperpolarization, and ROS bursts eventually resulting in cell death[10,48]. Here, we characterized an acetylation on K39 of Cytc in skeletal muscle which is gained after ischemia and was not detected under basal conditions, reversing the pattern typically seen with Cytc phosphorylation.

We here report the biological effect of K39 acetylation of Cytc with our Cytc purified from ischemic TA muscle, which shows a 58% increase in COX activity and a 45% decrease in downstream caspase-3 activity. We further characterized K39 acetylation using a site-directed mutagenesis approach. Previous work on Cytc has found glutamine substitution to be a suitable mimic for acetyl-lysine[22,23]. Additionally, using glutamine as the primary mimic for acetyl-lysine is an established technique for other proteins in general due to both side chains being polar but uncharged amides[31–34]. While lysine is conserved at residue 39 in mammals, glutamine is evolutionarily allowed in some species of plants and yeast[49]. Interestingly, in some experiments the glutamate substituted Cytc produced results comparable to those obtained with

the acetylmimetic glutamine, suggesting that other modifications of this residue, including the introduction of a negative charge via the glutamate side chain, on this highly evolutionary optimized protein can cause similar functional effects. We purified the mutant proteins after bacterial overexpression and generated Cytc double knockout stable cell lines overexpressing the mutant proteins to characterize the effects of the modification. In most experiments, the K39R mutant behaved similarly to WT, and the acetylmimetic K39Q and K39E mutants behaved similarly, indicating a switch-like behavior between positively charged lysine or arginine versus uncharged acetyl-mimic glutamine or negatively charged glutamate. While the glutamate mutant may behave similarly to the glutamine mutant in this publication, this is not generally true for other acetylation sites, and the glutamine should therefore be considered the acetyl-mimetic. It is worth noticing that Cytc purified from ischemic muscle exists as a mixture of acetylated and unacetylated protein, while acetylmimetic K39Q Cytc mimics 100% acetylation. This may explain the more pronounced effects seen using acetylmimetic K39Q protein vs. Cytc purified from ischemic muscle, such as for the caspase-3 activity assay. Additionally, the side chain of the K39R mutant features a delocalized positive charge, unlike that of lysine in the WT, and is spatially close to the propionate heme groups, which may explain some of the slight functional changes for K39R compared to WT seen in some assays. For example, OCR in the cell lines increased from WT to K39R to K39Q while ATP levels were similar for the K39R and K39Q cell lines, which

may be explained by slightly lower growth rate and thus lower ATP utilization for the K39R cell line (Supplementary Fig. 10B).

The reaction between Cyt*c* and COX is the proposed rate-limiting step of the electron transport chain[10], which is where >90% of cellular oxygen is consumed. However, the possibility that post-translational modifications of Cyt*c* can control overall flux of the ETC has previously been overlooked. The biochemical and cellular characterization of K39Q revealed that substitution of the lysine with glutamine stimulates respiration in vitro and in intact cells. A previous publication mutated K39 to leucine and did not find any effect on COX activity[50]. In contrast, in this study we saw profound increases in the reaction of COX with recombinant acetylmimetic K39Q Cyt*c* and K39 acetylated Cyt*c* isolated from ischemic porcine TA muscle. Additionally, intact cellular basal respiration of the K39Q cell line demonstrated a 70% increase in oxygen consumption compared to WT. The differences in COX activity seen in previous publications and here may be due to leucine being an uncharged, nonpolar residue while glutamine is an uncharged, polar residue. Interestingly, the NMR data helps explain the changes in functionality seen in this publication, despite K39 not being typically considered a part of the binding site with COX[10,50]. As seen with the electrostatic surface potentials and 2D $^1$H–$^{15}$N HSQC NMR CSPs, mutation at residue 39 modulates residues interacting with or near the heme group such as the M80-containing loop and the K55-to-W59 stretch. While the recombinant K39R protein did not result in large chemical shifts, the recombinant K39Q and K39E proteins displayed significant deviations from the WT. The changes seen in the NMR spectra (both CSPs and relaxation parameters), which affect the surroundings of the heme group, likely explain the increase in COX activity observed here by facilitating more efficient electron transfer from Cyt*c* to COX. Our previous work on Cyt*c* phosphorylation under basal conditions revealed an inhibition of respiration, maintaining an intermediate, optimal $\Delta\Psi_m$[11–20]. This protection was lost during ischemia which results in mitochondrial hyperpolarization upon reperfusion. In this study, we show that K39 acetylation, which is gained during ischemia, follows a different strategy to meet the energy demands of the cell by stimulating respiration.

Respiration drives the generation of the $\Delta\Psi_m$. Maintaining an intermediate, optimal $\Delta\Psi_m$ of 80 to 120 mV provides the full capacity to produce ATP while preventing excessive ROS generation[51,52]. When mitochondria are hyperpolarized, such as after ischemia-reperfusion, and $\Delta\Psi_m$ rises above 140 mV, ROS produced at complexes I and III increases exponentially[53]. The relationship between respiration, $\Delta\Psi_m$, and ROS is well supported by the data presented here. Under basal conditions, $\Delta\Psi_m$ of the K39Q cell line was higher than the WT. Similarly, the K39Q cell line showed higher mitochondrial ROS levels under basal conditions. Using our OGD/R model, we also assessed the effect of our acetylmimetic K39Q stable cell line on $\Delta\Psi_m$ and ROS after simulated ischemia-reperfusion. Interestingly, the K39Q stable cell line did not show a further increase in ROS after OGD/R, suggesting that the acetylmimetic replacement already supports higher $\Delta\Psi_m$ and ROS levels under basal conditions.

The purified recombinant K39Q protein demonstrated an interesting relationship with superoxide and $H_2O_2$. The acetylmimetic showed a reduced capability to scavenge superoxide, which is an important signaling molecule in normal muscle contraction[54–56]. Therefore, reduced scavenging of superoxide could be an adaptation to short periods of hypoxia that occur during normal muscle exercise. However, this response would become maladaptive in the context of ischemia-reperfusion injury where pathological superoxide levels mediate cellular damage. In contrast, the recombinant K39Q protein demonstrated an increased ability to scavenge $H_2O_2$.

Another major function of Cyt*c* is its involvement in apoptosis, where it performs multiple roles. When released from the mitochondrial IMS to the cytosol, Cyt*c* binds Apaf-1, activating the apoptosome and committing the cell to apoptosis. Previous publications strongly

suggested that K39 is a part of the Cyt*c*-Apaf-1 binding domain, where it is believed that Cyt*c* K39 forms a hydrogen bond with the carbonyl carbon of Apaf-1 phenylalanine 1063[3,45]. Our biochemical and cellular analysis of K39Q revealed that the substitution of the lysine with glutamine dramatically reduces the pro-apoptotic capability to activate downstream caspases. Additionally, the recombinant K39Q protein displayed reduced cardiolipin peroxidase activity and the K39Q cell line exhibited less cell death after exposure to $H_2O_2$, prolonged OGD/R, or ER stress. Furthermore, pro-apoptotic signaling was blocked in the K39Q cell line. Apaf-1 pull-down resulted in lower levels of K39Q Cyt*c* co-immunoprecipitation, and the K39Q cell line showed lower levels of caspase-9 and caspase-3 cleavage following induction of intrinsic apoptosis. These data further support a protective role for Cyt*c* K39 acetylation in skeletal muscle by reducing the ability of the protein to activate apoptosis.

Lysine acetylation is the most common post-translational modification in the mitochondria, particularly in the matrix where the basic pH and high concentrations of acetyl-CoA promote non-enzymatic acetylation[57,58]. In this publication, Cyt*c*, located in the IMS, was found to gain acetylation on K39 in tibialis anterior skeletal muscle as a response to ischemia. Porcine Cyt*c* has 18 lysines, which makes K39 acetylation a hypoxia-selective modification. This specificity adds to the likelihood that a specific acetyltransferase mediates Cyt*c* K39 acetylation. Interestingly, deletion of the acetyltransferase GCN5L1 in cardiomyocytes renders cells more sensitive to ischemia-reperfusion injury, with larger infarcts in the knockout hearts[59], and a previous publication found enrichment of GCN5L1 in the intermembrane space[60]. A second acetyltransferase candidate, ACAT1, was co-fractionated with Cyt*c* in a previous publication[61]. Elucidation of the specific acetyltransferase involved in the signaling pathway for this acetylation will be necessary to fully understand the role of this modification. Interestingly, we found that K39 acetylation was removed by sirtuin5 in vitro. Sirtuin5 has previously been shown to co-localize to the IMS with Cyt*c*[62], and Cyt*c* was previously suggested to be a substrate of sirtuin5-mediated deacetylation on an unknown site[63]. Noteworthy, sirtuins have been reported to be downregulated as a response to hypoxia[64], suggesting that blocking the removal of Cyt*c* K39 acetylation during ischemia facilitates the protective effect seen in skeletal muscle.

Previous work on quantitating lysine acetylation using mass spectrometry indicates that lysine acetylation occurs at very low stoichiometries[65]. It is known that the median stoichiometry for lysine acetylation, in general, is 0.02%, with highly acetylated proteins such as histones and acetyltransferase proteins being the only proteins to ever approach or even exceed 1% acetylation[66–69]. Despite this, lysine acetylation is known to be a highly biologically relevant modification[70]. By these standards, our results demonstrate that the ischemia-induced K39 acetylation of Cyt*c* is a more abundant acetylation, as detailed in Supplementary Table 2. To validate the specificity of K39 acetylation, we also used crystallography, and it is noteworthy that both mass spectroscopy and crystallography identified K39 acetylation only in ischemic, but not control, samples. The crystal structure of porcine K39 acetylated Cyt*c* at 1.50 Å resolution showed an average occupancy of 75% for the acetyl group, suggesting that a large proportion of the protein in the crystal structure is acetylated. Potential explanations for this discrepancy between the mass spectrometry and the crystallography occupancies may be that K39 acetylated Cyt*c* preferably crystallized within the ischemic TA sample, resulting in an over-representation of the acetyl group in the crystallography occupancy, or that some of the modification was lost during the sample preparation for mass spectrometry.

Overall, our findings support the assertion that K39 acetylation is protective during ischemia-reperfusion by enhancing energy production and at the same time disabling apoptotic capabilities. This acetylation may help explain the resilience of skeletal muscle to ischemia

compared to more sensitive tissues like the brain[28]. Skeletal muscle is a unique tissue type with drastic changes in perfusion depending on usage[71,72]. Cyt*c* K39 acetylation may be a helpful modification during exercise, allowing the muscle to meet the high energy demand while protecting the muscle from cell death. It is possible that this acetylation is a transient modification meant to handle short-term energy deficiencies. Additionally, individual muscles may have unique responses to ischemia. More research must be done to fully elucidate the signaling pathway controlling this acetylation. Identification of the pathway could aid in the development of therapeutic interventions for muscle ischemia-reperfusion injuries, such as during total knee arthroplasty, or chronic ischemic conditions, such as peripheral artery disease.

## Methods

### All chemicals and reagents were purchased from MilliporeSigma (Burlington, MA, USA) unless otherwise specified

**Tissue collection.** Porcine (domestic farm swine) tibialis anterior (TA) muscle samples from six animals (3 males and 3 females) were sourced from Wayne State University School of Medicine (Detroit, MI, USA) as discarded tissues with institutional animal care and use committee approval under protocol IACUC-18-11-0859. Tissues were rapidly harvested from animals post-euthanasia. For each animal, TA muscle from one hindlimb was immediately snap-frozen (control TA), while TA muscle from the other hindlimb was placed in a tightly sealed bag and incubated in a water bath at 37 °C for 1 h to induce ex vivo ischemia prior to being snap-frozen (ischemic TA), as we have previously done to induce ex vivo ischemia in brain tissue[19].

**Cytochrome c purification.** Cyt*c* was purified from non-ischemic control and ischemic TA tissue samples following our established protocol used on other tissues and organs in the past[11,13,19] which preserves post-translational modifications with minor adjustments. After homogenization, the tissue samples were subjected to acid extraction using 100 mM phosphate buffer, pH 3.9 at 4 °C. The pH of the mixture was adjusted to 4.0 using glacial acetic acid. After 10 min, the pH was adjusted to 4.3 using KOH and Cyt*c* was extracted overnight at 4 °C. The next day, samples underwent centrifugation at 24,000 x *g* for 45 min at 4 °C. The pH of the supernatants was adjusted to 6.5 and protease and phosphatase inhibitors were added to final concentrations of 1 mM phenylmethylsulfonyl fluoride (PMSF), 1 mM sodium vanadate, and 10 mM KF. The sample was then adjusted to pH 7.4 and underwent centrifugation as above. The conductivity of the resulting supernatants was adjusted to 3.0 mS/cm using ddH$_2$O and the supernatants were loaded onto a DE52-cellulose anion exchange column (Whatman; Piscataway, NJ, USA) equilibrated with 20 mM phosphate buffer, pH 7.4, with a conductivity of 3.0 mS/cm. The flowthrough of the column, which contains Cyt*c*, was adjusted to a conductivity of 5.0 mS/cm using 1 M phosphate buffer, pH 6.5, and to a pH of 6.5 using glacial acetic acid and loaded onto a CM52-cellulose cation exchange column (Whatman) equilibrated with 30 mM phosphate buffer, pH 6.5, and a conductivity of 5.0 mS/cm. The Cyt*c* bound on the CM52 column was oxidized using 2 mM potassium ferricyanide. The Cyt*c* fractions were then eluted using gradient elution from 50 mM to 150 mM phosphate buffer, pH 6.5, concentrated, and further purified via size exclusion chromatography using a Sephacryl S-100 gel filtration column (GE Healthcare; Chicago, IL, USA). Purified Cyt*c* samples were buffer exchanged to 20 mM ammonium bicarbonate, pH 7.6, using Amicon Ultra-15 10 kDa centrifugal filter units (#UFC901008, MilliporeSigma).

**Cytochrome c oxidase purification.** Cytochrome *c* oxidase (COX) was purified from porcine heart following our established protocol[19,73] which preserves post-translational modifications. Briefly, the porcine heart was homogenized, and the tissue extract was loaded onto a DEAE-Sepharose anion exchange column (GE Healthcare) equilibrated with 125 mM phosphate buffer and 0.5% Triton X-100, pH 7.4. The bound enzyme was eluted from the column using a salt gradient going from 100 mM to 700 mM phosphate buffer with 0.1% Triton X-100, pH 7.4. The enzyme was further purified via ammonium sulfate precipitation.

**Lysine 39 acetylation mapping by mass spectrometry.** Purified Cyt*c* (40 μg) was analyzed by mass spectrometry at the Max Planck Institute for Molecular Genetics (Berlin, Germany). Samples were initially processed by solubilization in 200 μL of 100 mM ammonium bicarbonate, pH 8, and boiled at 95 °C for 10 min on a rocking platform. Half of each sample was further processed via cysteine reduction by adding 5.5 mM tris(2-carboxyethyl)phosphine at 37 °C for 30 min at 800 rpm and via alkylation by adding 24 mM 2-chloroacetamide at room temperature for 30 min Each sample (pH 8.0) was split into two equal amounts and either digested by 200 ng trypsin or 200 ng chymotrypsin with shaking at 700 rpm at 37 °C for tryptic samples and 25 °C for chymotryptic samples overnight. To enhance enzyme activity, 4.5 μL of 100% acetonitrile was added to the tryptic samples. Peptide desalting was performed using Pierce C18 Spin Tips & Columns (#87784, Thermo Fisher Scientific; Waltham, MA, USA) according to the manufacturer's instructions. Desalted peptides were reconstituted in 5% acetonitrile and 2% formic acid in the water, briefly vortexed, and sonicated in a water bath for 5 min before injection to nano-LC-MS. Each run used 5 μg of digested protein.

LC-MS/MS was carried out by nanoflow reverse phase liquid chromatography (Dionex Ultimate 3000, Thermo Fisher Scientific) coupled online to a Q-Exactive HF Orbitrap mass spectrometer (Thermo Fisher Scientific), as previously described[74]. Briefly, the LC separation was performed using a PicoFrit analytical column (75 μm ID × 50 cm long, 15 μm Tip ID, New Objectives; Woburn, MA) in-house packed with 3 μm C18 resin (ReproSil-AQ Pur, Dr. Maisch; Ammerbuch, Germany). Peptides were eluted using a gradient from 3.8 to 38% solvent B (79.9% acetonitrile, 20% H$_2$O, and 0.1% formic acid) in solvent A (0.1 % formic acid) over 120 min at 266 nL/min flow rate. Nano-electrospray was generated by applying 3.5 kV. A cycle of one full Fourier transformation scan mass spectrum (300–1750 m/z, resolution of 60,000 at m/z 200, automatic gain control (AGC) target $1 \times 10^6$) was followed by 12 data-dependent MS/MS scans (resolution of 30,000, AGC target $5 \times 10^5$) with a normalized collision energy of 25 eV. To avoid repeated sequencing of the same peptides, a dynamic exclusion window of 30 sec was used. In addition, only peptide charge states between two to eight were sequenced.

Raw MS data were processed with MaxQuant software (v2.2.0.0) (Max Planck Institute of Biochemistry; Martinsried, Germany) and searched against the *Sus domesticus* proteome database UniProtKB with the ID UP000008227 (December 2016)[75]. Parameters of MaxQuant database searching were a false discovery rate (FDR) of 0.01 for proteins and peptides, a minimum peptide length of seven amino acids, a first search mass tolerance for peptides of 20 ppm, and a main search tolerance of 4.5 ppm. A maximum of two missed cleavages was allowed. Cysteine carbamidomethylation was set as a fixed modification, while N-terminal acetylation and methionine oxidation were set as variable modifications. Post-translational modifications of interest were lysine acetylation, mono-, di-, and tri-lysine methylation, as well as serine, threonine, and tyrosine phosphorylation, which were assessed via independent searches. All lysine acetylations of Cyt*c* were manually verified by the presence of at least one diagnostic peak (m/z 126 or 148 Da)[76]. The mass spectrometry data have been deposited to the ProteomeXchange Consortium (http://proteomecentral.proteomexchange.org) via the PRIDE[30] partner repository with the dataset identifier PXD040915. Note: In the raw mass spectrometry results, lysines were numbered including the start methionine (mature Cyt*c* lacks the start methionine), therefore the numbering will be one higher than what is reported here in the final manuscript text.

**Gel electrophoresis and western blotting.** Protein concentration was determined using the DC protein assay kit (#5000111, Bio-Rad; Hercules, CA, USA) according to the manufacturer's protocol. One μg of purified porcine TA muscle Cyt*c* of control and ischemic samples were run on a 10% tris-tricine SDS-PAGE gel, transferred onto immuno-blot PVDF membrane (#1620177, Bio-Rad) via wet transfer (140 mA, 40 min), and blocked in blocking reagent (5% non-fat dry milk in 1x TBS-T with 0.1% Tween 20) for 1 h at room temperature. For acetyl-lysine detection, the membrane was incubated with a 1:800 dilution of rabbit anti-acetyl-lysine conjugated to horseradish peroxidase secondary antibody (#6952S, Lot #2, Cell Signaling Technology; Danvers, Massachusetts, USA) in blocking reagent overnight at 4 °C. On a separate gel, one μg of the same samples were run, transferred, and blocked under identical conditions. This membrane was incubated with a 1:4000 dilution of mouse anti-Cyt*c* antibody (#556433, Lot #8213785, Clone #7H8.2C12, BD Pharmingen; San Jose, CA, USA) in blocking reagent overnight at 4 °C. The next day, the membrane for Cyt*c* was incubated with a 1:8000 dilution of sheep anti-mouse IgG conjugated to horseradish peroxidase secondary antibody (#NA931V, Lot #15273046, GE Healthcare) in blocking reagent for 2 h at room temperature. Blots were visualized using Pierce ECL western blotting substrate (#32106, Thermo Fisher Scientific).

Cyt*c* expression levels of cell lines stably expressing WT, K39R, K39Q, K39E, and negative control empty vector (EV) were assessed via western blot. Cells were lysed using 100 μL RIPA lysis buffer (150 mM NaCl, 5 mM EDTA, pH 8.0, 50 mM Tris, pH 8.0, 1% NP-40, 0.5% sodium deoxycholate, 0.1% sodium dodecyl sulfate (SDS)) supplemented with protease inhibitor cocktail (#P8340, MilliporeSigma), sonicated, and centrifuged at 16,900 x *g* for 20 min at 4 °C to remove cell debris. Sixty μg cell lysates were run on a 10% tris-tricine SDS-PAGE gel, transferred, and blocked as described above. The membrane was cut in half between the 25 and 35 kDa markers. The lower and upper membrane halves were incubated with a 1:1000 dilution of mouse anti-Cytc antibody or 1:5000 dilution of mouse anti-GAPDH antibody (#60004-1-Ig, Lot #10004129, Clone #1E6D9, Proteintech; Rosemont, IL, USA) as a loading control, respectively, in blocking reagent overnight at 4 °C. The next day, membrane halves were incubated with a 1:5000 or 1:10,000 dilution of sheep anti-mouse IgG conjugated to horseradish peroxidase secondary antibody in blocking reagent for 2 h at room temperature. The blots were visualized using Pierce ECL western blotting substrate.

Cell lines stably expressing WT, K39R, K39Q, K39E, and EV were assessed for PGC-1α, citrate synthase, and TFAM via western blot. Cells were lysed using 100 μL RIPA lysis buffer supplemented with protease inhibitor cocktail, sonicated, and centrifuged at 16,900 x *g* for 20 min at 4 °C to remove cell debris. Sixty μg cell lysates were run on a 10% tris-tricine SDS-PAGE gel, transferred, and blocked as described above. Membranes were probed using a 1:500 dilution of rabbit anti-PGC-1 antibody (#PA5-72948, Lot #XL3782595, Invitrogen; Carlsbad, CA, USA), a 1:1000 dilution of rabbit anti-citrate synthase antibody (#D7V8B, Lot #2, Cell Signaling Technology), a 1:2000 dilution of rabbit anti-tubulin antibody (#11224-1-AP, Lot #00016610, Proteintech) as a loading control, and a 1:1000 dilution of mouse anti-TFAM antibody (#MA5-16148, Lot #XL3781443, Clone #18G102B2E11, Invitrogen) overnight at 4 °C. The next day, the membranes were incubated with a 1:2500, 1:1000, or 1:2000 dilution of donkey anti-rabbit IgG conjugated to horseradish peroxidase secondary antibody (#NA934V, Lot #14879061, GE Healthcare), respectively, or a 1:5000 dilution of sheep anti-mouse IgG conjugated to horseradish peroxidase secondary antibody for 1 h at room temperature. The blots were visualized using Pierce ECL western blotting substrate.

**Immunoprecipitation (IP) and co-IP experiments.** For the TA muscle acetylation time course, TA muscle samples were snap-frozen (0 min ischemia) or were placed in a tightly sealed bag and incubated in a water bath at 37 °C for 15, 30, 45, or 60 min to induce ex vivo ischemia prior to being snap-frozen. Samples were lysed in IP buffer (50 mM Tris–HCl, pH 7.5, 150 mM NaCl, 1 mM EDTA, pH 8.0, 1% NP-40, 5 mM sodium pyrophosphate) supplemented with protease inhibitor cocktail. Additionally, the lysis buffer was also supplemented with 1 mM Trichostatin A (#T8552, MilliporeSigma) and 5 mM nicotinamide (#11127, Cayman Chemical; Ann Arbor, MI, USA) in order to inhibit acetylase and deacetylase enzymes. Lysed samples were sonicated, incubated for 30 min at 4 °C, and centrifuged at 16,900 x *g* for 20 min at 4 °C. Five mg cell lysates were incubated with mouse anti-Cyt*c* at 1 μg per mg of sample overnight at 4 °C. The next day, the mixtures were then incubated with Protein A/G PLUS-Agarose beads (#sc-2003, Santa Cruz Biotechnology; Dallas, TX, USA) overnight at 4 °C. Beads were washed and 40 μL of eluate from the beads were run on a 10% tris-tricine SDS-PAGE gel and transferred as described above. The membrane was blocked in BSA blocking reagent (5% BSA in 1x TBS-T with 0.1% Tween 20) for 1 h at room temperature. The membrane was incubated with a 1:1000 dilution of rabbit anti-acetyl-lysine conjugated to horseradish peroxidase antibody as described above in BSA blocking reagent overnight at 4 °C. On a separate gel, the eluate from the beads were run on a 10% tris-tricine SDS-PAGE gel, transferred, and blocked under identical conditions. As a loading control, the membrane was incubated with a 1:3000 dilution of mouse anti-Cyt*c* antibody in blocking reagent overnight at 4 °C. The next day, the membrane was incubated with a 1:8000 of dilution sheep anti-mouse IgG conjugated to horseradish peroxidase secondary antibody in blocking reagent for 1 h at room temperature. The blots were visualized using Pierce ECL western blotting substrate.

For the Apaf-1 Cyt*c* co-IP, Cyt*c* double knockout mouse embryonic fibroblasts stably expressing WT, K39R, K39Q, K39E Cyt*c* were utilized to assess the interaction between Apaf-1 and Cyt*c*. Cells (3 x 10^6) were seeded onto a 15 cm cell culture dish and cultured overnight. The next day, cells were washed twice with cold 1x PBS, harvested via scraping, and pelleted via centrifugation. The cell pellets were resuspended in 300 μL binding buffer (150 mM NaCl, 20 mM HEPES, pH 8.0, 10 mM KCl, 1.5 mM MgCl₂, 1 mM EDTA, 1 mM DTT, and 1 mM PMSF), lysed via sonication, and centrifuged at 16,900 x *g* for 20 min at 4 °C to remove cell debris. The supernatants containing Apaf-1 and Cyt*c* were incubated with 1 mM ATP for 30 min at 37 °C. For co-IP, 2 mg total protein from the above solutions were incubated with 4 μg rabbit anti-Apaf-1 antibody (#8969, Lot #2, Cell Signaling Technology) overnight at 4 °C. The next day, the mixtures were then incubated with Protein A/G PLUS-Agarose beads overnight at 4 °C. Beads were washed and 40 μL of eluate from the beads were run on a 10% tris-tricine SDS-PAGE gel, transferred, and blocked as described above. The membrane was cut in half between the 35 and 40 kDa markers. The lower and upper membrane halves were incubated with a 1:4000 dilution of mouse anti-Cyt*c* antibody or a 1:1000 dilution of rabbit anti-Apaf-1 antibody as a loading control, respectively, in blocking reagent overnight at 4 °C. The next day, membrane halves were incubated with a 1:8000 dilution of sheep anti-mouse IgG conjugated to horseradish peroxidase secondary antibody or a 1:8000 dilution of donkey anti-rabbit IgG conjugated to horseradish peroxidase secondary antibody, respectively, in blocking reagent for 1 h at room temperature. The blots were visualized using Pierce ECL western blotting substrate.

**Deacetylase assay.** Three μg of Cyt*c* from purified porcine TA muscle Cyt*c* of control and ischemic samples was performed in 50 mM Tris, pH 8.0, 4 mM MgCl₂, and 0.2 mM DTT at 37 °C for 3 h in the presence and absence of 1 mM NAD⁺ and 200 ng sirtuin5 (#S39-30H, SignalChem Biotech; Richmond, BC, Canada). The reaction was stopped via denaturation at 95 °C for 5 min. One and a half μg of each sample were run on a 10% tris-tricine SDS-PAGE gel, transferred, and blocked as described above. The membrane was probed using a 1:1000 dilution of rabbit anti-acetyl-lysine conjugated to horseradish peroxidase

antibody as described above. On a separate gel, 40 μL of eluate from the beads were run on a 10% tris-tricine SDS-PAGE gel, transferred, and blocked under identical conditions. As a loading control, the membrane was probed using a 1:3000 dilution of mouse anti-Cyt*c* antibody followed by a 1:8000 dilution of sheep anti-mouse IgG conjugated to horseradish peroxidase secondary antibody as described above. The blots were visualized using Pierce ECL western blotting substrate.

**Generation of recombinant Cytc plasmids via mutagenesis.** Somatic rodent Cyt*c* (WT) cDNA cloned into a bacterial pLW01 expression vector[77] was used to generate control K39R, acetylmimetic K39Q, and an additional control K39E Cyt*c* using the QuickChange Lightning site-directed mutagenesis kit (Agilent Technologies; Santa Clara, CA, USA) according to the manufacturer's protocol. The pLW01 plasmid also possesses the sequence for heme lyase, the enzyme necessary to covalently link heme and Cyt*c*, which bacteria lack. The following mutagenesis primers were used in amplification of each Cyt*c* plasmid: K39R forward primer: 5′-CTGTTTGGGCGGAGGACAGGCCAGGC–3′, K39R reverse primer: 5′-AGCCTGGCCTGTCCTCCGCCCAAACAG–3′, K39Q forward primer: 5′-TCTGTTTGGGCGGCAGACAGGCCAGGC–3′, K39Q reverse primer: 5′-GCCTGGCCTGTCTGCCGCCCAAACAGA–3′, K39E forward primer: 5′-TCTGTTTGGGCGGGAGACAGGCCAGGC–3′, K39E reverse primer: 5′-GCCTGGCCTGTCTCCCGCCCAAACAGA-3′. The PCR products were incubated for 1 h at 37 °C with DpnI endonuclease to remove methylated parental DNA and then transformed into XL 10-Gold Ultracompetent cells. Plasmids were purified from individual colonies using the Wizard Plus SV miniprep purification system (Promega; Madison, WI, USA). Purity and quantity were determined using a Nanodrop 1000 spectrophotometer (Thermo Fisher Scientific). The presence of the desired mutation in each plasmid was confirmed using DNA sequencing (Genewiz; South Plainfield, NJ, USA).

**Bacterial overexpression and purification of the recombinant proteins.** WT and mutant Cyt*c* constructs were transformed via heat shock into competent *E. coli* C41 (DE3) cells (Lucigen; Middleton, WI, USA) for protein overexpression. Transformed bacteria were cultured overnight at 37 °C in Terrific Broth medium (Research Products International; Mount Prospect, IL, USA) supplemented with 0.1 mg/mL carbenicillin. Once an optical density $(OD)_{600}$ of 0.8 was reached, Cyt*c* overexpression was induced by the addition of 100 μM isopropyl-β-D-thiogalactoside. After 6 h induction, bacterial cells were harvested and lysed, and Cyt*c* was purified as previously described[16]. Briefly, cells were pelleted via centrifugation at 8400 x *g* for 45 min at 4 °C, resuspended in 20 mM phosphate buffer, pH 7.4, supplemented with protease cocktail inhibitor, and then lysed via SLM Aminco French pressure system (American Instrument Co.; Silver Spring, MD, USA). Cell debris was removed via centrifugation at 26,200 x *g* for 45 min at 4 °C. Cyt*c* variants were purified via ion exchange chromatography as described above. Protein concentration was determined spectrophotometrically, while protein purity was determined via LabSafe GEL Blue (#786-35, G-Biosciences; St. Louis, MO, USA) staining of a 10% tris-tricine SDS-PAGE gel loaded with 500 ng of each Cyt*c* variant and ran as described above.

**Purified Cytc spectra and concentration determination.** The oxidized and reduced spectra of recombinant WT, K39R, K39Q, and K39E proteins were measured from 700 nm to 250 nm using a Jasco V-570 double beam spectrophotometer (JASCO Corporation; Hachioji, Tokyo, Japan). Additional oxidized spectra were taken from 750 nm to 620 nm to detect the characteristic heme peak at 695 nm. Cyt*c* was either fully oxidized with several grains of potassium ferricyanide $(K_3Fe(CN)_6)$ or fully reduced with several grains of sodium dithionite $(Na_2S_2O_4)$. Oxidized or reduced Cyt*c* proteins were run through NAP5 column (GE Healthcare) to separate the protein from the oxidizing or reducing agent. The concentration of Cyt*c* was calculated using the

absorbance at 550 nm of the oxidized and reduced spectra via the following formula: [Cyt*c*] in mM = (Abs550$_{reduced}$ − Abs550$_{oxidized}$) / (19.6 mM/cm x 1 cm) x dilution factor.

**Cytochrome c oxidase activity.** COX activities with the recombinant WT, K39R, K39Q, and K39E and the control and ischemic porcine TA muscle Cyt*c* were measured using purified regulatory-competent porcine heart COX. This is the same isozyme present in skeletal muscle. COX was diluted to 3 μM in solubilization buffer (10 mM K-HEPES, 40 mM KCl, 10 mM KF, 2 mM EGTA, 1% Tween 20, pH 7.4) supplemented with a 40-fold molar excess of tetraoleyl-cardiolipin (TOCL; MilliporeSigma) and 0.2 mM ATP. COX was dialyzed in 1 L solubilization buffer using a 12–14 kD Spectra/Por 2 dialysis membrane (#08-700-150, Spectrum Laboratories, Inc.; Rancho Dominguez, CA, USA) overnight at 4 °C to remove bound cholate[16]. The activity of COX (26.7 nM) in 220 μL solubilization buffer with recombinant Cyt*c* variants (0 – 25 μM) or control and ischemic porcine TA muscle Cyt*c* (5 μM) was measured using a Clark-type oxygen electrode (Oxygraph+, Hansatech; Pentney, UK) at 25 °C with 20 mM ascorbate as the electron donor. Oxygen consumption was recorded and analyzed using Oxytrace+ v1.0.48 software (Hansatech). The COX activity is reported as turnover number $(s^{-1})$.

**Caspase-3 activity.** The downstream caspase-3 activities after activation of Apaf-1 by recombinant WT, K39R, K39Q, and K39E and the control and ischemic porcine TA muscle Cyt*c* were measured using rhodamine fluorescence after caspase-3 mediated cleavage of Z-DEVD-R110 substrate. Cyt*c*$^{-/-}$ embryonic fibroblasts (CRL 2613, ATCC; Manassas, VA, USA) were cultured and lysed for isolation of cytosolic extracts as previously described[17]. Briefly, the knockout cells were cultured in 8 X 150-mm plates, trypsinized, pelleted, and washed twice with 1X PBS. Next, the pellets were washed once with a cold, hypotonic cytosolic extraction buffer (CEB; 20 mM K-HEPES, pH 7.5, 10 mM KCl, 1.5 mM MgCl$_2$, 1 mM EDTA, 1 mM EGTA, 1 mM dithiothreitol, 1 mM PMSF). The cell pellet was resuspended in 1 mL CEB and allowed to swell in the hypotonic CEB for 15 min at 4 °C. The suspension was transferred to a 2 mL Dounce homogenizer, and cells were disrupted via 30 to 35 strokes of the glass pestle. Cell breakage was monitored periodically by observing the homogenate on slides under a microscope. Lysates underwent centrifugation of 15,000 x *g* for 15 min at 4 °C to remove organelles and nuclei. The cytosolic extract was diluted to 2 mg/mL and incubated with 20 μg/mL of Cyt*c* for 2.5 h at 37 °C. The preincubated Cyt*c*-cytosolic extracts mixtures were utilized with the EnzChek Caspase-3 assay kit (#E13184, Invitrogen) according to the manufacturer's protocol with the caspase-3 inhibitor, Ac-DEVD-CHO, as a negative control. The reaction was initiated with the addition of the caspase-3 substrate, Z-DEVD-R110, a rhodamine-based tetrapeptide that fluoresces upon cleavage by caspase-3. Rhodamine fluorescence was measured using a Fluoroskan Ascent microplate reader (Labsystems, Thermo Fisher Scientific) with excitation/emission wavelengths of 485nm/527 nm excitation as previously described[17]. Caspase-3 activity is reported as a percentage of change in fluorescence (arbitrary units) compared to WT (recombinant Cyt*c* variant proteins) or control porcine TA muscle Cyt*c*.

**Rate of oxidation.** The kinetics of the redox reaction between WT, K39R, K39Q, and K39E and H$_2$O$_2$ were spectrophotometrically recorded to determine the rate of Cyt*c* oxidation by H$_2$O$_2$ as described[16,78]. Cyt*c* variants were fully reduced with Na$_2$S$_2$O$_4$ and subsequently desalted by passing through NAP5 columns, as described above. A cuvette was prepared containing 15 μM ferro-Cyt*c* and 0.2 M Tris-Cl, pH 7.0, and the baseline 550 nm absorbance was measured. An addition of 100 μM H$_2$O$_2$ initiated the reaction. The 550 nm absorbance was measured every 10 s for 2 min The decrease in the 550 nm absorbance corresponds to the oxidation of Cyt*c*. The initial rate of oxidation is reported in μM/s.

**Superoxide scavenging.** The kinetics of the redox reaction between WT, K39R, K39Q, and K39E Cyt$c$ and superoxide were analyzed spectrophotometrically using a hypoxanthine/xanthine oxidase reaction system[79,80] to determine the rate of superoxide scavenging by Cyt$c$. Cyt$c$ variants were fully oxidized with $K_3Fe(CN)_6$ and subsequently desalted by passing through NAP5 columns, as described above. A cuvette was prepared containing 10 μM ferri-Cyt$c$, 100 μM hypoxanthine, and 14.2 nM catalase (to prevent $H_2O_2$ accumulation) in 1x PBS. The production of superoxide was initiated with the addition of 181.5 nM xanthine oxidase. An addition of 925 nM superoxide dismutase 2 was also added for some experiments as a negative control that rapidly converts superoxide into $H_2O_2$. The 550 nm absorbance was remeasured every 15 s for 3 min The increase in the 550 nm absorbance corresponds to the amount of superoxide scavenged by Cyt$c$. The initial rate of reduction is reported in μM/s.

**Redox potential.** The midpoint redox potentials ($E^{o'}$) of WT, K39R, K39Q, and K39E Cyt$c$ variants were determined spectrophotometrically via the equilibration with a standard method[39] as previously described[17]. The reference compound was 2,6-dichloroindophenol (DCIP, $E^{o'} = 237$ mV), which has an absorption peak at 600 nm in the oxidized form. Briefly, 52.5 μM Cyt$c$, 15.87 μM $K_3Fe(CN)_6$, 31.75 mM citrate buffer, pH 6.5, and 31.75 μM DCIP were combined and baseline absorbances corresponding to fully oxidized Cyt$c$ (Abs550 – Abs570) and DCIP (Abs600) were recorded. Sequential additions of 1 μL of 5 mM ascorbate were added to reduce Cyt$c$, and the Abs550, Abs570, and Abs600 were recorded after each addition in one-minute intervals. At the end of the titration, a few grains of $Na_2S_2O_4$ were added to fully reduce Cyt$c$ and DCIP. In order to calculate $E^{o'}$, log (DCIP$_{ox}$/DCIP$_{red}$) was plotted against log (Cyt$c_{ox}$/Cyt$c_{red}$). This yielded a linear function with a slope, $n_{DCIP}/n_{Cytc}$, and a y-intercept, $n_{Cytc}/59.2(E_{Cytc}-E_{DCIP})$. Using the Nernst equation, these values were used to calculate $E^{o'}$ for each of the recombinant Cyt$c$ variants. $E^{o'}$ is reported as mV.

**Heme degradation.** The structural integrity of the covalently attached heme groups of WT, K39R, K39Q, and K39E Cyt$c$ was measured spectrophotometrically in the presence of high concentrations of $H_2O_2$[17,78]. Cyt$c$ was fully oxidized with $K_3Fe(CN)_6$ and desalted using NAP5 columns, as described above. A cuvette was prepared containing 5 μM ferri-Cyt$c$ in 50 mM phosphate buffer, pH 6.1, and the baseline 408 nm absorbance was measured. The degradation of heme was initiated with the addition of 3 mM $H_2O_2$. The 408 nm absorbance was measured at 60, 200, 400, 600, and 800 s. The decrease in the characteristic heme Soret band at 408 nm indicates heme degradation. The heme degradation is reported as a percentage of change in absorbance compared to the baseline absorbance.

**Cardiolipin peroxidase activity.** The cardiolipin peroxidase activities of WT, K39R, K39Q, and K39E Cyt$c$ were measured via resorufin formation, the oxidation product of Amplex red, as previously described[17,19,77]. Liposomes composed of 0%, 20%, 30%, and 50% of 18:1 tetraoleoyl-cardiolipin (TOCL, MilliporeSigma) with the remainder composed of 1,2-dioleoyl-sn-glycero-3-phosphocholine (DOPC, MilliporeSigma) were generated. Lyophilized TOCL and lyophilized DOPC were solubilized separately in chloroform to prepare working stock solutions of 10 mg/mL. The required volumes of the solubilized TOCL and DOPC to create different liposome compositions were mixed. The lipid mixtures were evaporated by blowing nitrogen gas on the liquid, resuspended in 20 mM K-HEPES, pH 7.2, and constituted into liposomes via sonication on ice for 5 x 30 s with 1 min intervals. Liposomes (25 μM) were incubated with 1 μM of the recombinant Cyt$c$ variant proteins at room temperature in a Costar 96-well plate (#3606, Corning Incorporated; Kennebunk, ME, USA) for 10 min to allow for equilibration of Cyt$c$-cardiolipin binding. Baseline resorufin fluorescence was measured using a Fluoroskan Ascent microplate reader with excitation/emission wavelengths of 530 nm/590 nm. The oxidation of cardiolipin was initiated with the addition of 10 μM Amplex red (#A36006, Invitrogen) and 5 μM $H_2O_2$. The fluorescence was remeasured at 60 and 300 s, during which the reaction rate was linear. Cardiolipin peroxidase activity is reported as fluorescence/minute (arbitrary units/min).

**Crystallography.** The recombinant K39Q Cyt$c$ and ischemic porcine TA muscle Cyt$c$ were buffers exchanged to water using Amicon Ultra-15 10 kDa centrifugal filter units (#UFC901008, MilliporeSigma). Samples were oxidized using 5 mM $K_3Fe(CN)_6$ immediately prior to crystallization. The approximate protein concentration for crystallization was 10-15 mg/mL for K39Q Cyt$c$ and 5-8 mg/mL for porcine K39 acetylated Cyt$c$. The sitting drop method was used with a 1 μL drop size protein plus 1 μL of reservoir solution. The best diffracting crystals of K39Q Cyt$c$ and ischemic porcine TA muscle Cyt$c$ grew to 0.2 mm x 1 mm size in Wizard Cryo2 screen #30 and 0.2 to 0.3 mm x 1.5 mm size in JBS screen #3A2, respectively (Supplementary Table 1). Crystals were flash frozen in liquid nitrogen directly from the drop (K39Q: 8DZL) or following cryoprotection with 20% $v/v$ ethylene glycol (porcine K39 acetylated Cyt$c$: 8DVX). Crystal diffraction data were collected at the Life Sciences Collaborative Access Team beamline 21-ID-D at the Advanced Photon Source, Argonne National Laboratory using a Dectris Eiger X 9M detector and integrated using the autoPROC program[81]. Both structures in Supplementary Table 1 were processed using the Phenix Program Suite: (1) solved using Phaser and rat native Cyt$c$ (5C0Z.pdb), (2) refit using the appropriate sequence in AutoBuild, and (3) refined using Phenix.Refine[82]. Both structures were also checked multiple times with PDB-REDO[83]. The refinement process was terminated when the two program suites converged to similar values of R-free. The fractional occupancies of the seven $K_3Fe(CN)_6$ molecules in 8DVX (Supplementary Table 1) were initially set to 0.5 and then refined by Phenix to values ranging from 0.65 to 0.97. The fractional occupancies of the four acetyl groups in 8DVX were initially set at 0.5 and then optimized with Phenix using a script kindly provided by Dr. Pavel Afonine that refined a separate, fractional occupancy for each acetyl group but left the occupancy of the associated 39 residues fixed at 1.0. Separate fractional occupancies were used for the four acetyl groups to account for the different mobilities of the acetyl-lysine residues.

The four molecules in the crystal structure of K39Q (8DZL) were superposed onto molecule A in the rat WT structure (5C0Za) using YASARA version 20-07-04[84]. The five superposed structures were then loaded into Molecule Operating Environment (MOE, Chemical Computing Group; Montreal, Canada)[85] and all residues with at least one atom within 3.5 Å of Gln39 in molecule A (8DZLa) were displayed. The comparable figure for Aly39 (8DVX) was prepared using the same protocol.

The RMSD values, relative to molecule A in the crystal structure of WT rodent Cyt$c$ (5C0Za)[19], were calculated for molecule A in the crystal structures of T28E (5DF5a)[16], K39Q (8DZLa), S47E (6N1Oa)[19], and K53Q (7LJXa)[22], using YASARA version 20-07-04[84] (Analyze > "individual objects on selected" > Atoms). The RMSD plot for the backbone atoms was generated with Microsoft Excel using data imported from YASARA.

**Electrostatic surface potential.** The structures used to calculate the electrostatic surface potential were based on the crystallographic structure of mouse Cyt$c$ (Protein Data Bank, PDB, ID: 5C0Z) as previously described[23]. All reduced (WT, K39R, K39Q, and K39E) structural models were built using the WT protein as a template with UCSF Chimera 1.15 software (University of California; San Francisco, CA, USA)[86]. The force field parameters for the reduced heme group[87] were used to generate the topology and coordinate files, required for the simulation, with the TLEAP module of AmberTools 2021 (University

of California)[88]. Each structure was summited to 1 μs MD simulation prior to the calculation of the electrostatic surface potential with UCSF Chimera 1.15 software (University of California).

**Protein nuclear magnetic resonance analyses.** *E. coli* C41 (DE3) cells were transformed with WT, K39R, K39Q, and K39E mutant pLW01-Cyt*c* constructs and were plated on LB-agar plates containing 0.1 mg/mL ampicillin. Single colonies were incubated at 37 °C with 150 rpm agitation for 16 h. For the NMR experiments, the overnight culture was transferred to 1 L of minimal M9 medium supplemented with 16.7 μg/mL δ-aminolevulinic acid hydrochloride and $^{15}NH_4Cl$ as the sole nitrogen source. Protein expression was induced with 1 mM isopropyl-β-D-thiogalactoside at an $OD_{600}$ of 0.8, and the cultures were incubated for 20 h to allow for protein expression. Cells were harvested by centrifugation at 9000 x *g* for 10 min at 4 °C, suspended in lysis buffer (10 mM Tricine-NaOH, pH 8.5, 1 mM PMSF, 0.02 mg/mL DNase supplemented with cOmplete protease inhibitor cocktail (#11836145001, Roche; Mannheim, Germany)), and physically ruptured by sonication. Cellular debris was separated by centrifugation at 14,000 x *g* for 30 min at 4 °C. Protein purification was carried out by ion chromatography with a Nuvia-S column (#7324720, Bio-Rad) using a Fast Protein Liquid Chromatography (FPLC) system (Bio-Rad). Protein purity and sample homogeneity were determined spectrophotometrically and by running 5 μg of each protein sample on a 15% SDS-PAGE, followed by Coomassie blue staining. Dynamic light scattering and tryptic digestion analyses confirmed the monomeric state of the protein samples and the point mutations of the Cyt*c* species.

Protein samples were dialyzed against 10 mM sodium phosphate buffer, pH 6.5, for NMR experiments, which were performed on a 700 MHz Bruker Avance-III (Bruker; Billerica, Massachusetts, USA) and 500 MHz Bruker Avance-III (Bruker) spectrometers equipped with a cryoprobe. Samples contained 800 μM of reduced $^{15}N$-labelled Cyt*c* in 10 mM sodium phosphate buffer, pH 6.5, 5% $D_2O$. Sodium ascorbate was used to ensure the redox state of the sample. Backbone amide group resonances of the recombinant Cyt*c* proteins were monitored by recording 2D $^{15}N$-$^{1}H$ Heteronuclear Single Quantum Correlation (HSQC) experiments as previously described[23]. In order to assign the backbone resonances, additional 3D $^{15}N$-$^{1}H$ TOtal Correlation SpectroscopY (TOCSY)-HSQC and 3D $^{15}N$-$^{1}H$ Nuclear Overhauser Effect Spectroscopy (NOESY)-HSQC experiments were performed. The experimental data were processed using Bruker TopSpin NMR 4.1.1 software (Bruker), and assignment and chemical shift perturbations (CSPs) were analyzed with NMRFAM-Sparky 1.470 NMR assignment tool (University of Wisconsin-Madison; Madison, WI, USA)[89]. The average difference (each difference calculated as follows: $\Delta X = X_{K39Q} - X_{WT}$) in $^{15}N$ relaxation parameters (longitudinal relaxation rate ($R_1$), transversal relaxation rate ($R_2$), and heteronuclear $^{15}N\{^{1}H\}$ (NOE)) were obtained from experiments recorded at 500 MHz $^{1}H$ frequency at 25 °C for the recombinant WT and K39Q proteins. Each difference was calculated from a set of relaxation parameters measured on two biologically independent samples for WT and K39Q Cyt*c*. All attempts at replication were successful. The $R_1$ parameter was calculated using 7 delays ranging from 10 to 1000 ms, and the $R_2$ parameter was calculated using 7 delays ranging from 16.96 to 118.72 ms. The $^{15}N\{^{1}H\}$ NOE was determined by recording spectra with and without a 4.2 s long proton saturation period. The spectra were processed using NMRPipe 10.9 and analyzed with NMRDraw 10.9[90]. Peak intensities were extracted from each spectrum, with R1 and R2 parameteres for each protein sample obtained by fitting the data to an exponential decay function: $I_t = I_0 e^{(-tR_{1,2})}$, where $I_0$ represents the peak intensity at time 0 and $I_t$ represents the intensity at time t. The $R_2/R_1$ ratio was used to estimate the rotational correlation time ($\tau_c$) of the protein variants using TENSOR 2.0 and ROTDIF 1.1 software[91,92].

**Stable expression of Cyt*c* variants in mammalian cells.** Somatic rodent Cyt*c* (WT) cDNA was cloned into the pBABE-puromycin expression plasmid (Addgene; Cambridge, MA, USA) between the BamHI and EcoRI restriction sites with the following outer primers: forward primer 5′-CATGGGTGATGTTGAAAAAGGCGAGAAGATTTTTG TTCAAAAGTG−3′ and reverse primer 5′-CACTTTTGAACAAAAATCTTC TCGCCTTTTTCAACATCACCCATG−3′[16]. A similar PCR mutagenesis approach and mutagenesis primers as above were used to generate WT, K39R, K39Q, K39E, and negative control empty vector (EV) constructs, which were then transfected into Cyt*c* double knockout mouse embryonic lung fibroblast cells[41] (kind gift from Dr. Carlos Moraes, University of Miami) using Transfast transfection reagent (#E2431, Promega). The transfected cells were cultured in growth media (DMEM (#11965-092, Gibco; Grand Island, NY, USA), 10% FBS (#16000-044, Gibco), 100 μg/mL primocin (#ant-pm-1, Invivogen; San Diego, CA, USA), 1 mM sodium pyruvate (#11360-070, Gibco), 50 μg/mL uridine (#U3003, MilliporeSigma)) at 37 °C with 5% $CO_2$, unless otherwise stated. Stable cell lines expressing the four recombinant Cyt*c* variants and EV were selected in the presence of 2 μg/mL puromycin.

**Growth rate.** The growth rate of the stable cell lines expressing WT, K39R, K39Q, K39E Cyt*c*, and EV were measured. Cells were seeded at 30,000 cells/well onto a 0.1% gelatin coated Costar 24-well plate (#3524, Corning Incorporated) and cultured in 2 mL/well growth media. Every 24 h, cells were washed with 1x PBS and collected via trypsinization. Collected cells were counted using a Moxi Z mini automated cell counter (#MXZ001, Orflo Technologies; Ketchum, ID, USA).

**ATP production.** The basal ATP levels of the stable cell lines expressing WT, K39R, K39Q, K39E Cyt*c*, and EV were measured. Cells ($1 \times 10^6$) were seeded onto a 10 cm cell culture dish (#664160, Greiner Bio-One; Frickenhausen, Germany) and cultured overnight. The next day, in under 90 s, the cells were washed once with 1x PBS, harvested via scraping, pelleted via centrifugation, and flash frozen in liquid nitrogen. The pellets were stored at −80 °C until analysis. To lyse the cells, the samples were boiled for 2 min in 300 μL boiling buffer (100 mM Tris-Cl, 4 mM EDTA, pH 7.75) and sonicated on ice. Samples were diluted 300-fold and 40 μL of the diluted samples were used to determine the ATP concentration using the ATP bioluminescence assay kit HS II (#11699709001, Roche) according to the manufacturer's protocol with an Optocomp I luminometer (MGM Instruments; Hamden, CT, USA). The ATP level is reported as μmol ATP/mg protein.

**Mitochondrial stress test.** The oxygen consumption rates (OCR) and extracellular acidification rates (ECAR) of the stable cell lines expressing WT, K39R, K39Q, K39E Cyt*c*, and EV were measured. Cells were seeded at 25,000 cells/well onto a 0.1% gelatin coated Seahorse XF$^e$24 cell culture microplate (#100777-004, Agilent Technologies) and cultured overnight in 250 μL/well growth media. The growth media was replaced with 675 μL/well seahorse media generated by dissolving 4.15 g DMEM powder (#D5030, MilliporeSigma) in 500 mL ddH$_2$O, pH 7.4, sterile filtered (#431097, Corning Incorporated), and supplemented with 10 mM glucose and 10 mM sodium pyruvate without phenol red, FBS, or sodium bicarbonate. After the media change, cells were incubated in a $CO_2$ free incubator for 1 h, and then OCR and ECAR were measured in an XF$^e$24 Seahorse extracellular flux analyzer according to manufacturer's protocol. Mitochondrial stress test was performed via sequential injection of 1 μM oligomycin, 2.5 μM carbonylcyanide-3-chlorophenylhydrazone (CCCP), and 1 μM rotenone/antimycin A. Basal respiration, non-mitochondrial oxygen consumption, proton leak, ATP-coupled respiration, maximal respiration, and spare respiratory capacity were calculated according to manufacturer's protocol. Briefly, non-mitochondrial oxygen consumption was calculated using the OCR values after the final injection of 1 μM rotenone and antimycin A, which inhibit complexes I

and III, respectively. The basal respiration was calculated using the initial OCR values prior to any chemical injections minus the non-mitochondrial oxygen consumption. The proton leak was calculated using the OCR values after the injection of 1 μM oligomycin, which inhibits ATP synthase, minus the non-mitochondrial oxygen consumption. ATP-coupled respiration was calculated using basal respiration minus the proton leak. Maximal respiration was calculated using the OCR values after the injection of 2.5 μM CCCP, which is a mitochondrial uncoupler, minus the non-mitochondrial oxygen consumption. The spare respiratory capacity was calculated using maximal respiration minus the basal respiration. OCR and ECAR were normalized to total protein content in the well after the experiment to control for variation in cell number between cell lines. For post-mitochondrial stress test normalization, cells in each well were lysed using 25 μL RIPA lysis buffer (see above) supplemented with protease inhibitor cocktail and protein concentration was determined by DC protein assay kit (#5000111, Bio-Rad). OCR is reported as pmol $O_2$/minute/μg protein and the ECAR is reported as mpH/minute/μg protein.

**Mitochondrial membrane potential.** The relative mitochondrial membrane potentials ($\Delta\Psi_m$) of the stable cell lines expressing WT, K39R, K39Q, K39E, and EV were measured using the JC-10 probe (#ENZ-52305, Enzo Life Sciences). Cells were seeded at 25,000 cells/well onto 0.1% gelatin coated Costar 96-well plates and cultured overnight in 100 μL/well growth media. The next day, cells were washed with 1x PBS and incubated for 30 min at 37 °C with 3 μM JC-10 in DMEM (#31053-028, Gibco) without phenol red or FBS. After incubation, cells were washed twice with 1x PBS. JC-10 can selectively enter mitochondria where it exists as a monomer at low $\Delta\Psi_m$ with primarily green fluorescence (excitation/emission of 485 nm/527 nm). When $\Delta\Psi_m$ is high, JC-10 aggregates with primarily red fluorescence (excitation/emission of 485 nm/590 nm). Fluorescence was measured in PBS using a Synergy H1 microplate reader (BioTek Instruments Inc.; Winooski, VT, USA). $\Delta\Psi_m$ is reported as the ratio of red/green fluorescence.

**Mitochondrial ROS.** The production of mitochondrial ROS of the stable cell lines expressing WT, K39R, K39Q, K39E Cyt*c*, and EV were measured using MitoSOX (#M36008, Invitrogen). MitoSOX can selectively enter mitochondria where it fluoresces red (excitation/emission of 510 nm/580 nm) after reaction with superoxide. Cells were seeded at 25,000 cells/well onto 0.1% gelatin coated Costar 96-well plates and cultured overnight in growth media. The next day, cells were washed with 1x PBS and incubated for 30 min at 37 °C with 5 μM mitoSOX in DMEM (#31053-028, Gibco) without phenol red or FBS. After incubation, cells were washed twice with 1x PBS and fluorescence was measured in PBS using a Synergy H1 microplate reader. Mitochondrial ROS production is reported as a percentage of change in fluorescence/μg protein (arbitrary units/μg protein) compared to WT.

**Oxygen-glucose deprivation/reoxygenation (OGD/R) experiments.** To simulate ischemia/reperfusion injury in a cell culture model, stable cell lines expressing WT, K39R, K39Q, K39E Cyt*c*, and EV were exposed to transient oxygen-glucose deprivation followed by reoxygenation (OGD/R) as previously described[20]. Cells were seeded and cultured according to the above protocols for JC-10 or MitoSOX. After culturing overnight, the growth media was exchanged for ischemia-mimetic DMEM media (#A14430-01, Gibco) without glucose, phenol red, or FBS that had been bubbled with 95% $N_2$ and 5% $CO_2$. Cells were incubated in ischemia-mimetic media at 37 °C for 90 min with 1% $O_2$ and 5% $CO_2$ in a hypoxia chamber controlled by ProOx 110 Compact $O_2$ Controller (RRID: SCR_021129, BioSpherix; Redfield, NY, USA) and ProCO2 120 Compact $CO_2$ Controller (RRID: SCR_021127, BioSpherix). After ischemia, cells were reoxygenated via incubation in glucose-containing DMEM (#31053-028, Gibco) without phenol red or FBS, containing the respective probe, at 37 °C for 30 min with 5% $CO_2$. The JC-10 or

MitoSOX experiments were then carried out as described in the methods above.

**Cell death using annexin V/propidium iodide staining.** Cell death levels of the stable cell lines expressing WT, K39R, K39Q, K39E Cyt*c*, and EV were quantified after exposure to $H_2O_2$ or OGD/R using annexin V and propidium iodide (PI) staining. Annexin V is impermeable to the intact plasma membrane and binds phosphatidylserine, a lipid normally present on the inner leaflet of the plasma membrane which flips to the outer leaflet early during apoptosis. Therefore, annexin V is commonly used as a marker of early apoptotic cells. PI is also impermeable to the intact plasma membrane and is a DNA intercalator. During necrosis, holes in the plasma membrane allow PI to enter. Therefore, PI is commonly used as a marker of necrotic cells[93]. Cells that stain double positive for both annexin V and PI are late apoptotic cells. Cells ($1 \times 10^6$) were seeded on 10 cm cell culture dishes and cultured overnight in growth media. The next day, cells were exposed to either $H_2O_2$ (400 μM for 16 h), OGD/R (16 h oxygen-glucose deprivation/1 h reoxygenation) as described above, or thapsigargin (#328570050, Thermo Fisher Scientific) (1 mM for 24 h). After 48 h from the initial seeding, cells were trypsinized, washed twice with 1x PBS, counted, and a total of $1 \times 10^6$ cells were resuspended in 1 mL 1x annexin V binding buffer from the FITC annexin V apoptosis detection kit I (#556547; RRID: AB_2869082; BD Pharmingen). A total of 450 μL cell suspension was incubated for 15 min with 6 μL annexin V and 6 μL PI. Data were collected using a CyFlow Space flow cytometer (Sysmex America, Inc.; Lincolnshire, IL, USA). The results were analyzed using FCS Express 7 software (De Novo Software; Glendale, CA, USA).

**Caspase cleavage.** Caspase cleavage of the cell lines stably expressing WT, K39R, K39Q, K39E, and EV were assessed via western blot. Cells ($3 \times 10^6$) were seeded onto 15 cm cell culture dishes (#CC76823614, USA Scientific; Ocala, FL, USA) and cultured overnight. The next day, cells were treated with 1 μM staurosporine (#NC0662740, Enzo Life Science; Farmingdale, NY, USA) for 8 h, which is known to be a potent inducer of intrinsic apoptosis[17,46,47]. Cells were washed with 1x PBS, harvested via trypsinization, and pelleted via centrifugation. The cell pellets were resuspended and lysed in 200 μL RIPA lysis buffer supplemented with protease inhibitor cocktail, sonicated, and centrifuged at 16,900 x $g$ for 20 min at 4 °C to remove cell debris. Ninety μg cell lysates were run on a 10% tris-tricine SDS-PAGE gel, transferred, and blocked as described above. Membranes were incubated with either a 1:500 dilution of mouse anti-caspase-9 antibody (#9508S, Lot #7, Cell Signaling Technology) or a 1:1000 dilution of rabbit anti-caspase-3 antibody (#14220S, Lot #4, Cell Signaling Technology) in blocking reagent overnight at 4 °C. The next day, the membrane was incubated with a 1:8000 dilution of sheep anti-mouse IgG conjugated to horseradish peroxidase secondary antibody or a 1:8000 dilution of donkey anti-rabbit IgG conjugated to horseradish peroxidase secondary antibody, respectively, for 1 h at room temperature. The blots were visualized using Pierce ECL western blotting substrate. The next night, the membranes were incubated with a 1:5000 dilution of mouse anti-GAPDH antibody, as a loading control, in blocking reagent overnight at 4 °C. The next day, the membrane was incubated with a 1:8000 dilution of sheep anti-mouse IgG conjugated to horseradish peroxidase secondary antibody in blocking reagent for 1 h at room temperature. The blots were visualized using Pierce ECL western blotting substrate.

**Statistical analyses.** Data shown represent the mean, and error bars represent the standard deviation between independent trials for protein experiments or individual wells/cell culture dishes for cellular experiments. Statistical analyses of the data were performed using Graphpad Prism v9.4.1 (Graphpad Software; San Diego, CA, USA). Most data were analyzed using one-way ANOVA comparing the mean of each column with the mean of the control column (WT) with the Dunnett

post-hoc test unless otherwise specified. COX activity of the recombinant purified proteins was performed using one-way ANOVA as described above on the 25 µM Cyt*c* condition. Heme degradation of the recombinant purified proteins was performed using one-way ANOVA as described above on the 800 s condition. For annexin V/PI experiments, a one-way ANOVA as described above was performed to compare total cell death (PI+ cells, annexin V+ cells, and annexin V+/PI + cells combined). A student's two-tailed t test assuming equal variance was used for COX activity (to compare porcine TA muscle samples), caspase-3 activity (to compare porcine TA muscle samples), and OGD/R (to compare normoxia and hypoxia results within each cell line). P values are indicated in the figure legends. The experiments were not randomized, and investigators were not blinded to allocation during experiments or outcome assessment.

### Reporting summary
Further information on research design is available in the Nature Portfolio Reporting Summary linked to this article.

### Data availability
The mass spectrometry data generated in this study have been deposited in the PRIDE repository under accession code PXD040915. The *Sus domesticus* proteome database UniProtKB with the ID UP000008227 and 26,104 entries (December 2016) was searched. The crystallography data generated in this study are available in the PDB repository with the identifiers 8DZL and 8DVX. The remaining data generated in this study are provided in the Supplementary Information/Source Data file. Source data are provided with this paper.

### Code availability
The Phenix.Refine script is available in Supplementary Figure 1A.

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

## Acknowledgements
This work was supported by the National Institutes of Health grants R01 GM116807 (M.H.) and R01 NS120322 (M.H.), a Competitive Graduate Research Assistant Award from the Wayne State University Graduate School (P.T.M), and research grants from the Spanish Ministry of Science and Innovation (PGC2018-096049-B-I00 and PID2021-126663NB-I00; I.D-M.), European Regional Development Fund (FEDER; I.D-M.), Andalusian Government (BIO-198, US-1254317, US-1257019, P18-FR-3487, and P18HO-4091, US/JUNTA/FEDER, UE; I.D-M), University of Seville (VI PPIT) and the Ramón Areces Foundation (I.D-M.). G.P-M. was awarded a Ph.D. fellowship from the Spanish Ministry of Education, Culture, and Sport (FPU17/04604). This research used resources from the Advanced Photon Source, a U.S. Department of Energy Office of Science User Facility operated for the Department of Energy Office of Science by Argonne National Laboratory under Contract DE-AC02- 06CH11357. Use of the Life Sciences Collaborative Access Team (LS-CAT) Sector 21 was supported by the Michigan Economic Development Corp., the Michigan Technology Tri-Corridor (Grant 085P1000817; M.H.), and Wayne State University's Office of the Vice-President for Research (M.H.). We thank Dr. Carlos Moraes at the University of Miami for providing the Cytc double knockout cell line, Beata Lukaszewska-McGreal for proteome sample preparation, and Dr. Pavel Afonine for an alternate script for Phenix. refine, Philippe Archambault of the Chemical Computing Group for his technical advice, the support of the Max Planck Society, and the staff from the NMR facility at CITIUS (University of Seville).

## Author contributions
P.T.M., G.P-M., J.W., A.A.T., I.M., H.A.K., A.V., M.P.Z., P.P.H., K.K., T.A., M.A.R., D.D.C., I.L., M.H.M, T.H.S., B.F.P.E., I.D-M., and M.H. designed and performed experiments. D.M. conducted mass spectrometry experiments. A.V., J.S.B., and B.F.P.E. determined the crystal structures. P.T.M., A.A.T., G.P-M., B.F.P.E., I.D-M., and M.H. wrote the manuscript. All authors have read and agreed to the final version of the manuscript.

## Competing interests
The authors declare no competing interests.
