## [Peer review file · Nature Communications]

REVIEWER COMMENTS

Reviewer #1 (Remarks to the Author):

Morse, Turner, Pérez-Mejías, et al. report a novel lysine acetylation site on cytochrome C, K39, that occurs after ischemia in porcine tibialis anterior muscle. The authors then investigate the functional repercussions of this modification through several in vitro assays and structural characterization. Using an acetyl lysine mimetic via site-directed mutagenesis, K39Q, to investigate how modified CytC might alter apoptosis, mitochondrial respiration, and ROS scavenging. Ultimately, they report this novel acetyl lysine site to be protective during ischemia-reperfusion by enhancing energy production and at the same time disabling apoptotic capabilities. I was asked to review this manuscript to provide perspective on the mass spectrometry data, which needs further detail prior to publication, and I also provide other comments below.

1.) Mass spectrometry was used mainly as the discovery tool to find where acetylation was occurring, and then also to confirm site-directed mutagenesis. It is not the focus of the manuscript, nor do I think it needs to be. That said several questions about the mass spec data should be addressed in the manuscript:

1a) How many spectra were identified with the acetyl lysine modification? Was the false discovery rate calculated separately for modified and unmodified spectra? Do these spectra contain the ammonium ion common to acK (m/z 126, doi.org/10.1021/ac800005n, PMID: 18338905,) which would aid confidence in identifications? How prevalent is this acK ammonium ion in the data set? Are there other acetyl lysine sites that exist, and how many acetyl lysine modifications were localized to K39 versus a different lysine residue?

1b) Can the authors comment on occupancy using the mass spec data? They report an occupancy of 75% with the NMR data. This is quite high, especially for reversible modifications like acetyl lysine, which the authors suggest in the discussion might be the case with Cytoc acK39. If it is indeed that high, it should also be relatively straightforward to estimate from the LC-MS/MS data by comparing modified and non-modified peptide abundances. Is that occupancy a reason that enrichment for acetylated peptides was not needed, too? Also, the occupancy looks far less than this in the western blots shown in Figure 1b. Why is the western blot occupancy so different from estimates with structural techniques? Including brief reference to these points in the discussion about occupancy would be useful.

1c) The manuscript starts by saying that acK39 was discovered from purified Cytoc, but the reason that acetylation was the only modification considered was not discussed. How did the authors decide to

look for acetylation? Was phosphorylation considered at all in this experiment given its relevance in previous work on Cytoc in ischemia? Were there any phosphosite identified?

1d) The raw data and results files should be posted publicly in a repository like ProteomeXchange.

2.) Even with the referenced text, more details about the mass spectrometry sample preparation, data acquisition methods, and data processing should be included in this manuscript so that readers do not have to chase down that information. Furthermore, what software was used for Supp Figure 4?

3.) To support the discussion of the possible role of acetyl K39 as a transient protective modification, the authors should investigate K39 deacetylation by a panel of sirtuins, especially given CytoC's mitochondrial importance. Fully delineating the acetyl transferase and deacetylation dynamics will require more, as the authors suggest, but their hypothesis would be strengthened in this manuscript by commenting on whether this modification can be removed by the known acetyl lysine erasers that would make it a transient protective modification during ischemia.

Reviewer #2 (Remarks to the Author):

This is highly interesting study describing the impact of post-translational modifications on various mitochondrial functions and respiration. The authors have identified a new acetylation site at lysine 39 in ischemic porcine skeletal muscle. Their findings clearly demonstrate that lysine 39 acetylation is an adaptive response in skeletal muscle to meet heightened energy demand and provide robust resilience to ischemia-reperfusion injury in tissues. The presented data support the conclusions drawn in this manuscript. There are some weaknesses that need to be addressed to further increase the importance of these findings.

The abstract could be expanded to include more key findings of this exciting impactful research.

The purpose of performing experiments related to molecular dynamics simulations is not clear, the authors may want to elaborate more on this or provide evidence for the requirements of molecular dynamics simulation in context of cytochrome c acetylation in tissues.

The authors have mentioned that the substitution of the lysine with glutamine dramatically reduces the pro-apoptotic capability to activate downstream caspases. The authors may provide evidence that this modification destabilizes the cytochrome c interaction with Apaf-1 using cell lysates or using recombinant mutant cytochrome c, which ultimately leads to the blockade of apoptosome formation and caspase activation.

The authors have shown that K39Q and K39E reduce caspase activity. Caspase-9 and -3 cleavage using western blot may also be shown to further establish that caspases were not cleaved leading to the inhibition of their activity.

Do K39Q and K39E modulate mitochondrial biogenesis or affect mitochondrial DNA or membrane potential modifying pathways?

Figure 7 may further be expanded to include more data related to the reduction/inhibition of cell death in response to other types of stress.

Reviewer #3 (Remarks to the Author):

Key results

The authors have employed X-ray crystallography, NMR and various assays to show that K39 acetylation on Cytochrome C makes skeletal muscles resistant to ischemia reperfusion injury. Higher levels of mitochondrial potential and ROS in the acetyl mimetic in basal condition is attributed as the reason for ischemia reperfusion injury resistance. K39Q has an improved peroxide but reduced superoxide scavenging, it reduces cell apoptosis via reduced apaf1 binding. Mass spec data shows acetylation on K39 when cells had undergone ischemia vs in its absence. Heme is an important component of CytC and was used to ensure that all mutants, namely K39Q (acetyl-mimetic), K39R (non-acetylated lysine mimetic) and K39E (lysine with a negative charge mimetic) retained the coordination with Heme. The authors then explore the role of acetylation in cell death, studied using interaction between Apaf-1 and K39, rate of oxidation and reduction in acetylated and non-acetylated K39 mimetic. Overall, the study shows that acetylation on K39 is a novel finding with role in regulation of CytC.

Validity

The authors have in sufficient and accurate detail explained the data and derived conclusions.

Significance

CytC is an essential part of ETC and its role in respiration and cell death makes it necessary to understand the regulation. Authors have given a broad overview of how previous literature discusses the role of phosphorylation and acetylation in regulation and even cancer. Results from this study inform the field about the novel K39 site implicated in regulation.

Data and methodology

The authors have used multiple techniques and assays to capture the effects of acetylation in Lys39. The results have been described thoroughly, appropriate controls and statistical analysis has been conducted for each assay. The choice of assays is inclusive for the type of parameters that need to be studied, to validate the effect of K39 acetylation.

Analytical approach

The statistical analysis in the paper to highlight results of significance is satisfactory.

Suggested improvements

Improvements in Table 1:

->Unit cell dimensions can be split in two rows and only report to one digit after decimal, like such (similar to the Table from validation report):

34.4, 52.5, 61.6

110.0, 92.8, 92.0

->In CRYSTAL DATA:

Resolution high-low should be in a single row, like such- 49.22-1.12 (xx-xx)

Resolution info should also have the resolution range of the highest/best resolution shell (see above in parenthesis). This is done for 'completeness' and 'average I/sigma' but not for resolution)

The resolution-high and low is repeated, unsure why this is repeated?

Missing units for wavelength

In Table 1 for acetylated porcine CytC, the well condition is "10% PEG4K, 20% isopropanol, 15% PEG 4K, pH 5.0". This has PEG4k twice, which one is the correct % of PEG4k? The documentation for JBS 3 Screen on the company website shows no buffer in the condition, was this condition adjusted for pH to 5.0 (as stated in Table 1)?

Was there any anomalous signal from Fe in Heme?

->Electron density in PDB 8DVX, Chain A for ALY39 is not obvious. The acetyl group has been assigned an occupancy of 37%. The methods section mentions a script that was used to assign this occupancy. The omit map shown in Figure 3e does not give much information if placing an acetyl group is justified. Please comment on the following:

Was the omit map created removing the acetyl group or the entire lysine39?

What was the lowest occupancy that the refinement was started at for this group?

Was there difference density if ONLY acetyl group is removed in this position (Chain A, LYS39)?

->In Fig 6b, basal levels of ATP content are similar for K39Q and K39R, how does this affect the conclusions that are derived from increased ATP coupled respiration in acetyl mimetic (K39Q)?

->Since heme is important for proper folding of CytC, in case of recombinant CytC purification, it's not obvious from the method section how heme was supplemented. At what step was heme added to the bacterial cells?

->Placement of HOH 421 and HOH 399 in Chain D of 4DXL is not very convincing. Have the authors tried modeling an alt conf of Gln16 and refining?

->Both K39E and K39Q look to be good acetyl mimetics (K39E has more dramatic effects in some assays). Higher focus seems to be on K39Q in the paper overall. For related future studies, would authors consider both K39E and K39Q to be equally good mimetics?

Clarity and context

The language is clear and thorough. The authors could potentially talk more about the crystal structures that already exist with either of these modifications (K39 acetylation) or other modifications (such as phosphorylation).

Reviewer #4 (Remarks to the Author):

In the current study, Morse et al. describe a novel acetylation on residue K39 in CytC during ischemia that confers altered protein function; specifically, increased oxidase activity and decreased apoptosis. Based on these results, the authors claim that acetylation could be a protective mechanism during ischemia and help tissue meet high energy demands.

Understanding the effects of post-translational modifications on the function of CytC with a key role in the respiratory electron transport chain and apoptosis is of high importance. Overall, the data presented in this work supports a change in the function of CytC by K39 acetylation. The main concern is that the data presented for the K39Q mutant and other mutants (e.g., K39E) clearly indicate that the former is not a better mimetic of acetylation. It is as good as or comparable to the other mutants. There are other important concerns that need to be addressed. Please, see comments below:

1. Experimental controls on tissue treatments seem to be lacking. No information is provided to determine whether acetylation is statistically significant. I suggest the following:

- Analyze by mass spectrometry and immunoblotting a larger number of tissue samples. For example, 12 tissue samples (2 from each hindlimb of three animals, 4 samples per animal) and report the results for all samples as supplementary information.

- The work is partially funded by NIH, which requires sex to be considered as a biological variable in studies involving vertebrate animals. The sex of the animals has not been considered. At least, please indicate the sex of the animals used.

- The process described in the methods section to induce ischemia involves keeping the tissue at 37 °C for 1 h. There is not reference provided in the manuscript to determine whether this is an acceptable protocol. Please, test the presence of acetylation as a function of time within the 1 h period and report the results.

2. The labels and numbers in Figure 1a are barely visible. Please, increase font size and provide a better explanation in the figure of the results that need to be highlighted. Please, indicate how many mass spec. were obtained and how many immunoblotting gels were performed.

3. No clear justification is provided of why the K39Q mutant is a suitable mutation to represent acetylation.

4. Figure 1d (not the inset) seems to show only three proteins: WT, K39E and K39R. If K39Q, which is the selected mutant is overlapping, please indicate so and indicate with which curve it is overlapping.

5. There is a significant difference between the caspase-3 activity of the K39Q mutant (90% decrease in activity) compared with the naturally acetylated version of the induced—ischemic sample (45% decrease activity). Although, the trend is in the right direction, these results imply that there are important differences between the mutant and the acetylated wildtype. The sentences in lines 109 and 100 need to indicate this significant difference to consider the suitability of the K39Q mutant.

6. Based on Figures 1 and 2, it is clear that K39 has important effects on CytC function. However, the K39Q mutation does not seem to be a good mimetic. Data presented in Figures 1g, 2a and 2b, show that both mutants K39Q and K39E have similar effects, even though E is negatively charged, and Q is neutral as the acetylated K. Thus, references throughout the manuscript indicating that the K39Q mutant is a better mimetic are somewhat misleading, as well as the titles of the figure legends specifically highlighting this mutant.

7. The symbols in Figures 1 and 2 (circle, inverted triangle, etc.) are very small, making it very hard for the reader to identify differences between them.

8. The importance of the MD simulations does not seem to be clearly explained. The MD results show that the 20-30 loop in the K39 acetyl-lysine is more flexible than in the other structures. Is there an explanation for this result in relation to the claim of the manuscript?

9. In relation to the crystallographic results, I suggest representing the potential changes in the structure of the backbone and side chains in the regions nearby K39Q and the acetylated K39 in the 3D space to facilitate the identification of the differences between them.

10. The chemical shift perturbation data indicate that mutations in K39 by either E or Q led to structural changes (more pronounced for the less conservative mutation) in the regions close to amino acids 57 and 80. The mutation K39R is not perturbing the native structure significantly. However, this latter mutation results in modified function(e.g., Fig. 6b, c , d). Could the authors provide an explanation in the manuscript about this apparent contradiction?

11. The differences in relaxation data between the WT protein and the K39Q mutant cannot be interpreted as presented because there are no errors associated to the measurements or at least the errors are not mentioned in the manuscript. By reading the methods section, it seems that the R1, R2 and het-NOE values were obtained once. Some of the differences observed could be related to error measurement. NMR relaxation experiments must be acquired at least twice to be able to report error/deviation.

12. The results on the effect of K39Q in the increase of ATP levels compared to WT and the basal oxygen consumption rate are similar to the other mutant K39R and K39E. These results are analogous to data shown in Figure 1 and 2, where we can conclude that K39 is an important amino acid controlling these functions, but the K39Q is as good mimetic as K39E and K39R in some instances.

Introduction:

We, the authors, would like to extend our thanks for the thoughtful and thorough critiques, which we have addressed below. We believe that the corrections and experiments requested by the reviewers have significantly strengthened the quality of the revised manuscript.

Responses to the reviewers' comments:

Reviewer 1

1. How many spectra were identified with the acetyl lysine modification? Was the false discovery rate calculated separately for modified and unmodified spectra? Do these spectra contain the immonium ion common to acK (m/z 126, doi.org/10.1021/ac800005n, PMID: 18338905,) which would aid confidence in identifications? How prevalent is this acK immonium ion in the data set? Are there other acetyl lysine sites that exist, and how many acetyl lysine modifications were localized to K39 versus a different lysine residue?

In the revised manuscript, we tested 6 control porcine TA samples (3 male and 3 female) and 6 ischemic porcine TA samples (3 male and 3 female). The K39 acetylation was not identified in any of the control TA samples, but was identified in 3 of the ischemic TA samples (1 male and 2 female). The false discovery rate was always 1% and is calculated independently for protein-level- and PTM-level false discovery rates in the MaxQuant program. Yes, diagnostic IMacK ions (mass 126 and 148 Da) were found in all ischemic TA samples that possessed K39 acetylation. See the below table:

		Control TA Samples						Ischemic TA Samples					
		1	2	3	4	5	6	1	2	3	4	5	6
K39 acetylation	126 Da								X		X		
	148 Da							X			X		

X indicates that the respective diagnostic peak was present

We did identify other lysine acetylations in some of our 12 samples, in addition to K39 acetylation. However, we would like to note that these acetylations were present in both the control and ischemic samples in roughly equal numbers. On the other hand, K39 acetylation was never present in the control conditions and was gained during ischemia, making it ischemia-specific and the focus of this current manuscript. Specifically, these other acetylations were on lysine 27, 79, and 86 within CytC. See the below table:

Acetylation Site	Control TA Samples						Ischemic TA Samples						
	1	2	3	4	5	6	1	2	3	4	5	6	
K27	X		X					X		X	X		
K39							X	X		X			
K79				X						X			
K86	X		X							X	X		

X indicates that the respective acetylation site was present

These other acetylations are less interesting because they are present in skeletal muscle under basal conditions and remain during ischemia. We also identified K88 acetylation, but the intensity of the fragments was so low that an occupancy could not be calculated, therefore we have elected not to include

it in the above table. While the focus of this current manuscript is on K39 acetylation due to it being an adaptive response to ischemia, we plan to study these other acetylations in follow up publications. Note: In the raw mass spectrometry results, lysines were numbered including the start methionine (mature CytC lacks the start methionine), therefore the numbering will be one number higher than what is reported in the final manuscript text and this rebuttal letter. The above has all been clarified in the revised manuscript.

2. Can the authors comment on occupancy using the mass spec data? They report an occupancy of 75%. This is quite high, especially for reversible modifications like acetyl lysine, which the authors suggest in the discussion might be the case with CytC acK39. If it is indeed that high, it should also be relatively straightforward to estimate from the LC-MS/MS data by comparing modified and non-modified peptide abundances. Is that occupancy a reason that enrichment for acetylated peptides was not needed, too? Also, the occupancy looks far less than this in the western blots shown in Figure 1b. Why is the western blot occupancy so different from estimates with structural techniques? Including brief reference to these points in the discussion about occupancy would be useful.

Previous work on quantitating lysine acetylation using mass spectrometry indicates that lysine acetylation occurs at very low stoichiometries (PMID: 24187339). It is known that the median stoichiometry for lysine acetylation is 0.02%, with highly acetylated proteins such as histones and acetyltransferase proteins being the only proteins to ever approach or even exceed 1% acetylation (PMID: 30837475, PMID: 31296352, PMID: 24489116, PMID: 26839187). Despite this, lysine acetylation is known to be a highly biologically relevant post-translational modification. As for our results, the mass spectrometry gave occupancies for K39 acetylation varying from 0.83% to 0.04% (see Supplemental Table 1). This is on the high end for lysine acetylation occupancies identified by mass spectrometry and is comparable to that identified on other highly acetylated proteins.

On the other hand, the crystal structure showed a much higher occupancies of ~75%. It is possible that the crystal structure overestimates the occupancy due to the acetyl-lysine somehow selectively crystallizing over the non-acetylated protein. The revised discussion section has been significantly expanded regarding the topic of occupancy.

Regarding the western blot in Figure 1B, it was performed using two different antibodies, anti-CytC and anti-acetyl-lysine. These antibodies have different binding affinities to their designated epitopes, and the blots were processed using different concentrations of each antibody; therefore, a fair comparison of signal cannot be made between an anti-CytC band vs. an anti-acetyl-lysine band.

3. The manuscript starts by saying that acK39 was discovered from purified CytC, but the reason that acetylation was the only modification considered was not discussed. How did the authors decided to look for acetylation? Was phosphorylation considered at all in this experiment given its relevance in previous work on CytC in ischemia? Were there any phosphosite identified?

Originally, we were expecting and looking for phosphorylation sites, which is why our CytC purification protocol includes phosphatase inhibitors (notably, we did not include any deacetylase inhibitors). The discovery of this novel acetylation site was totally unexpected. When we analyze our samples by mass spectrometry, we always search for phosphorylation, acetylation, and mono-/di-/tri-methylation. No enrichment strategy was used for any post-translational modification. We have added a small comment to the beginning of the discussion highlighting the serendipitous nature of this discovery. Surprisingly, no phosphorylations of CytC or any other type of post-translational modification of CytC were identified in any spectra included in this manuscript: only lysine acetylation was identified. It should be noted that this is our first study to examine CytC in tibialis anterior skeletal muscle or any skeletal muscle, a tissue that is very different compared to other tissues in which we and others have reported phosphorylation of CytC. The current findings underpin the tissue-specificity of CytC regulation, which we propose allows fine-

tuning of its multiple life-sustaining and pro-apoptosis functions in tissues and organs specialized to certain functions, energy requirements, and sensitivity to apoptosis.

4. The raw data and results files should be posted publicly in a repository like ProteomeXchange.

We have completed this, as requested. The accession code is PXD040915 and the DOI is 10.6019/PXD040915. Reviewer access may be found via the following information:

<https://www.ebi.ac.uk/pride/login>

Username: reviewer_pxd040915@ebi.ac.uk

Password: 2l4PJkZ

The proteome data will be publicly available upon acceptance of the manuscript.

5. Even with the referenced text, more details about the mass spectrometry sample preparation, data acquisition methods, and data processing should be included in this manuscript so that readers do not have to chase down that information. Furthermore, what software was used for Supp Figure 4?

We have significantly expanded the revised methods for the mass spectrometry section and the requested information is now provided.

For revised Supplemental Figure 4, data was collected using MALDI-TOF Ultraflextreme system (Bruker) configured on positive linear mode (samples in solution) and positive reflectron mode (tryptic digestion samples). The figure panels were made with Origin 2019b.

6. To support the discussion of the possible role of acetyl K39 as a transient protective modification, the authors should investigate K39 deacetylation by a panel of sirtuins, especially given Cytc's mitochondrial importance. Fully delineating the acetyl transferase and deacetylation dynamics will require more, as the authors suggest, but their hypothesis would be strengthened in this manuscript by commenting on whether this modification can be removed by the known acetyl lysine erasers that would make it a transient protective modification during ischemia.

We thank the reviewer for suggesting this experiment. We have identified that K39 acetylation is removed in vitro by Sirtuin5. We have included these data in the revised Figure 1D. This is particularly interesting because Sirtuins are known to be downregulated as a response to hypoxia, providing a potential mechanistic explanation for K39 acetylation and the protective effects seen during ischemia. We have expanded the revised discussion section to incorporate these results.

Reviewer 2

1. The abstract could be expanded to include more key findings of this exciting, impactful research.

We appreciate the comment. While there are word limits regarding the abstract that are set by the journal, we have done our best to highlight how these results could explain skeletal muscle resilience to ischemia-reperfusion injury in the revised abstract.

2. The purpose of performing experiments related to molecular dynamics simulations is not clear, the authors may want to elaborate more on this or provide evidence for the requirements of molecular dynamics simulation in context of cytochrome c acetylation in tissues.

Upon a critical review of the manuscript, we agree with the general theme of multiple comments by the reviewers that the molecular dynamics simulations in the original submission do not contribute significantly to the overall story of the manuscript and that the purpose of these experiments was unclear. At first, we sought to answer this feedback by performing new molecular dynamics simulations using different force field parameters. Our goal was to find force field parameters that generated molecular dynamics results that identified regions of mobility similar to those identified in the NMR dataset. Identifying molecular dynamics force field parameters that produce results similar to the NMR would be useful in that it could inform future studies about how best to simulate Cytc. While we were able to somewhat improve the agreement between the molecular dynamics simulations and the NMR data with these new simulations, there were still large differences in the results. Therefore, we have elected to remove the molecular dynamics section entirely in order to streamline the revised manuscript. We feel that removing this and instead focusing on actual experimental data improves the overall quality of the manuscript.

3. The authors have mentioned that the substitution of the lysine with glutamine dramatically reduces the pro-apoptotic capability to activate downstream caspases. The authors may provide evidence that this modification destabilizes the cytochrome c interaction with Apaf-1 using cell lysates or using recombinant mutant cytochrome c, which ultimately leads to the blockade of apoptosome formation and caspase activation.

We thank the reviewer for suggesting this experiment. As requested, we have performed Apaf-1 immunoprecipitation experiments on our cell lysates and measured the level of Cytc that co-immunoprecipitated. Importantly, high levels of Cytc were pulled down in the WT and K39R cell lines, while lower levels of Cytc were pulled down in the K39Q and K39E cell lines. This directly illustrates that the interaction between Apaf-1 and K39Q/K39E Cytc is impaired. We have included these data in the revised Figure 7D.

4. The authors have shown that K39Q and K39E reduce caspase activity. Caspase-9 and -3 cleavage using western blot may also be shown to further establish that caspases were not cleaved leading to the inhibition of their activity.

We thank the reviewer for suggesting this experiment. As requested, we have performed western blots on our cell lysates looking for caspase-9 and caspase-3 cleavage after treatment with 8 h of staurosporine, a potent inducer of intrinsic apoptosis. In support of our model, we saw lower levels of caspase-9 and caspase-3 cleavage in the K39Q and K39E cell lines compared to the WT and K39R cell lines. This further supports that the interaction between Apaf-1 and K39Q/K39E Cytc is impaired, as we can immediately see impaired cleavage of these downstream components of the caspase cascade. We have included these data in the revised Figure 7E, 7F.

5. Do K39Q and K39E modulate mitochondrial biogenesis or affect mitochondrial DNA or membrane potential modifying pathways?

As requested, we have performed western blot analyses on our cell lysates looking to assess the levels of PGC-1 α , citrate synthase, and TFAM as markers for mitochondrial biogenesis, mitochondrial mass, and mitochondrial DNA synthesis. We did not see a significant difference between the WT, K39R, K39Q, K39E, and EV cell lines. We have included these data in the revised Supplemental Figure 11A and in the revised results.

We do see some changes in mitochondrial membrane potential comparing cells expressing the WT vs. K39Q/K39E, which we previously measured via the JC-10 probe. This increase in mitochondrial membrane potential is explained by the increased mitochondrial activity (i.e., increased proton pumping creating a large voltage differential). These data are shown in Figure 6E and 6G.

6. Figure 7 may further be expanded to include more data related to the reduction/inhibition of cell death in response to other types of stress.

To expand Figure 7, we tested how our cell lines responded to ER stress via thapsigargin treatment. The results were consistent with the levels of cell death shown after exposure to H₂O₂ and OGD/R: the cell line expressing the K39Q Cytc showed reduced levels of cell death compared to the cell line expressing the WT Cytc. We have included these data in the revised Figure 7C. We have also expanded on how the cell line expressing K39Q interferes with pro-apoptotic signaling. We have included these data in the revised Figure 7D-7F.

Reviewer 3

1. Improvements in Table 1:
 - a. Unit cell dimensions can be split in two rows and only report to one digit after decimal, like such (similar to the Table from validation report):
 - b. 34.4, 52.5, 61.6
 - c. 110.0, 92.8, 92.0

We have completed this, as requested.

2. Improvements for CRYSTAL DATA:
 - a. Resolution high-low should be in a single row, like such- 49.22-1.12 (xx-xx)
 - b. Resolution info should also have the resolution range of the highest/best resolution shell (see above in parenthesis). This is done for 'completeness' and 'average I/sigma' but not for resolution)
 - c. The resolution-high and low is repeated, unsure why this is repeated?
 - d. Missing units for wavelength

Thank you for catching the above errors. We have corrected them, as requested.

3. In Table 1 for acetylated porcine Cyt_c, the well condition is "10% PEG4K, 20% isopropanol, 15% PEG 4K, pH 5.0". This has PEG4k twice, which one is the correct % of PEG4k? The documentation for JBS 3 Screen on the company website shows no buffer in the condition, was this condition adjusted for pH to 5.0 (as stated in Table 1)?

The PEG 4K was 10% in the original crystallization solution but was increased to 15% in the cryoprotectant. We have deleted the erroneous PEG 4K within the table. Crystallization condition JBS 3A2 has no buffer. However, its pH measured in our laboratory was 5.0.

4. Was there any anomalous signal from Fe in Heme?

Anomalous diffraction data were collected from the Aly39 and K39Q crystals. When analyzed using the Coot software, the anomalous diffraction peaks for the four heme iron atoms in the Aly39 structure (8DVX) ranged from 21.4 to 20.2 RMSD and the 7 peaks from the FC6 molecules ranged from 16.1 RMSD for FC6 301/B, with a refined occupancy of 94%, to 8.0 RMSD for FC6 601/B, with a refined occupancy of 65%. The highest peak for a sulfur atom was 8.1 RMSD for Met80/B. The highest peak for a sulfur atom in the K39Q structure (8DZL), which has no FC6 molecules, was 7.4 RMSD for Cys17/A.

5. Electron density in PDB 8DVX, Chain A for ALY39 is not obvious. The acetyl group has been assigned an occupancy of 37%. The methods section mentions a script that was used to assign this occupancy. The omit map shown in Figure 3e does not give much information if placing an acetyl group is justified. Please comment on the following:
 - a. Was the omit map created removing the acetyl group or the entire lysine39?
 - b. What was the lowest occupancy that the refinement was started at for this group?
 - c. Was there a different density if ONLY acetyl group is removed in this position (Chain A, LYS39)?

The default Phenix simple: omit map is assembled by sequentially removing 5% of the residues per submap that are then combined into the full omit map.

The starting occupancy for the deposited structure was 0.5 for all 4 acetyl-lysine 39 (Aly39). If all four Aly39 occupancies in 8DVX are set to 0.1, and re-refined, they approach the published values (0.37 / 0.81 / 0.95 / 0.89) after 6 cycles of occupancy refinement with Phenix. Refine (0.35 / 0.79 / 0.94 / 0.88).

As a control calculation, Lys53, which is acetylated in prostate cancer cells (see PMID: 33916826), but not in ischemic muscle, was converted to Aly53 in all 4 molecules and refined using the same Phenix script and data as the Aly39 structure to evaluate the contribution of background electron density to the refined occupancies of the acetyl group. The four refined occupancies for the acetyl group of Aly53 ranged from 0.09 to 0.33, which are all lower than the occupancies for the Aly39 structure.

We believe that molecule A is acetylated but shows lower density for several reasons. The main reason is that molecule A favored crystallizing non-acetylated Lys39 over Aly39 in contrast to molecules B, C, and D where Aly39 appears to have preferentially crystallized. To empirically test this, we replaced Aly39 with Lys39 in molecule A and simulated the hydrogen bond interactions. This resulted in better hydrogen bonding to Glu62 in a symmetry-related Cytc molecule when using Lys39 (-23.4 kcal/mol) compared to when Aly39 was used (-5.5 kcal/mol). See below diagrams with Lys39 on the left and Aly39 on the right.

This simulation was performed in MOE. The crystal contacts within 10 Å of the 4 Cytc molecules in the crystal asymmetric unit were generated and added to the crystal structure in MOE. MOE then looked for any crystal contacts within 4.5 Å of the four Aly39 residues. Only molecule A had any crystal contacts. Glu62 is from a symmetry generated molecule C, and Glu4 is from a symmetry generated molecule A. These two glutamates were allowed to move to optimize any possible interactions with the residues in the crystal contact.

This stronger interaction with lysine selectively favors Lys39 over Aly39 in molecule A during crystallization. As a result, we see the lower density for molecule A. Of course, there is still some Aly39 at molecule A site within the actual crystal, but it is discouraged at this position due to the crystal packing. Additionally, the density is further lowered because molecule A has the biggest volume for the residue to move around within, therefore what electron density from Aly39 is present is then diffused over a larger volume compared to the other three sites.

We hope that this answers the question regarding why molecule A has a lower occupancy for the acetyl group compared to molecules B, C, and D.

6. In Fig 6b, basal levels of ATP content are similar for K39Q and K39R, how does this affect the conclusions that are derived from increased ATP coupled respiration in acetyl mimetic (K39Q)?

The K39R cell line tends to demonstrate a hybrid phenotype, usually falling between the WT and K39Q for most assays. Based on the basal OCR and the ATP-coupled respiration being higher for the K39Q cell

line, we anticipated that the K39Q cell line would also demonstrate higher levels of ATP than the K39R cell line. Therefore, it is interesting that the ATP content is instead similar between the K39R and K39Q cell lines. We propose that this is due to the growth rate of the K39R cell line being slightly slower than the WT, K39Q, and K39E cell lines (see revised Supplemental Figure 11B). As the K39R cell line is dividing slightly slower, it is consuming slightly less energy and therefore has a slightly higher basal ATP than otherwise expected, thus accounting for the slight discrepancy. We have included this in the revised discussion.

7. Since heme is important for proper folding of Cyt_c, in case of recombinant Cyt_c purification, it's not obvious from the method section how heme was supplemented. At what step was heme added to the bacterial cells?

Bacteria produce heme, so heme supplementation was not necessary. However, one key difference between bacteria and mammalian cells regarding heme is that bacteria do not have the heme lyase enzyme, which covalently attaches heme to Cyt_c. Therefore, our pLW01 bacterial expression vectors were engineered to contain both the sequence for the respective modified Cyt_c as well as the sequence for heme lyase. In this manner, the heme is correctly and covalently incorporated into Cyt_c within the bacteria, as shown by our spectroscopy results. The presence of the heme lyase has been pointed out in the revised methods section.

8. Placement of HOH 421 and HOH 399 in Chain D of 8DZL is not very convincing. Have the authors tried modeling an alt conf of Gln16 and refining?

When analyzed with COOT, the original conformation (#1) is near to but does not match rotamer#1 for glutamine. The alternative conformation defined by the electron density for the two waters (#2) is near to but does not match the orientation of rotamer#3 for glutamine. For both refined conformations, COOT flags the bonds, angles, and chirals as outliers (colored red). However, the values for all three parameters are slightly better for the original conformation. Conformation#1 has robust 2Fo-Fc density and conforms better to the conformations of Gln16 in the three other chains. Also, almost every program that might use the 8DZL structure – e.g., for molecular dynamics – will only import the first alternative conformation. Consequently, we see no pressing need to add the second conformation to the deposited structure.

9. Both K39E and K39Q look to be good acetyl mimetics (K39E has more dramatic effects in some assays). Higher focus seems to be on K39Q in the paper overall. For related future studies, would authors consider both K39E and K39Q to be equally good mimetics?

Using glutamine as a mimetic for acetyl-lysine is an established technique due to their similar side chains, which are both uncharged amides (PMID: 19875076, PMID: 28901832). Therefore, K39Q is the acetylmimetic. This has been clarified in the revised results and discussion sections upon the first mention of the acetylmimetic approach. Glutamate is not a good mimetic for acetyl-lysine because it has a negatively charged carboxylic acid side chain, which is not reflective of the charge or functional group of acetyl-lysine.

Independently of this, the K39 residue of Cyt_c demonstrates switch-like behavior whether the residue is positively charged vs. uncharged/negatively charged. While it seems to be the case that both K39Q and K39E Cyt_c demonstrate similar behavior for K39 of Cyt_c, this does not hold true for other residues of Cyt_c or other proteins in general. Anecdotally, we can share that for other acetylation sites of Cyt_c, the glutamate mutant does not behave similarly to the glutamine mutant or the in vivo acetylated proteins (unpublished). Given the above explanation, we would not consider K39E to be an acetylmimetic.

The K39E mutant was included to test the hypothesis that change in charge drives maximal functional effects. Given that lysine is positively charged, glutamine is neutral but polar like acetyl-lysine, and

glutamine is negatively charged, the K39Q and K39E replacements produce a change of 1 and 2 charge units compared to WT, respectively. As predicted, K39E Cytc shows a more pronounced effect to activate COX activity compared to K39Q and WT, producing the trend: WT < K39Q < K39E Cytc (Figure 1G).

10. The authors could potentially talk more about the crystal structures that already exist with either of these modifications (K39 acetylation) or other modifications (such as phosphorylation).

As requested, we have calculated the RMSD values comparing acetylmimetic (K39Q and K53Q) and phosphomimetic (T28E and S47E) mutations of Cytc against the native, rodent WT structure. These data are available in Supplemental Figure 1C. Interestingly, acetylmimetic mutations (K39Q and K53Q) tend to more globally perturb the entire Cytc structure compared to phosphomimetic mutations (T28E and S47E) which show much fewer differences compared to WT. Specifically, both the K39Q and K53Q structures have variation in the D50-to-G60 and N70-to-E90 regions, while the T28E and S47E structures have greatly reduced variability compared to the WT in these regions and globally, highlighting that acetylation may have stronger effects on Cytc function than phosphorylation.

Reviewer 4

1. The main concern is that the data presented for the K39Q mutant and other mutants (e.g., K39E) clearly indicate that the former is not a better mimetic of acetylation. It is as good as or comparable to the other mutants.

We appreciate the reviewer's concern. However, using lysine to glutamine replacement is an effective and generally accepted mimetic for acetyl-lysine due to their similar side chains, which are both uncharged amides and often closely mimic the functional effects seen with lysine-acetylated proteins (PMID: 17938198; PMID: 19875076, PMID: 34455704; PMID: 34753925; PMID: 28901832). Additional references have been included in the manuscript to show that glutamine is generally accepted as a model for lysine acetylation. Therefore, K39Q is the acetylmimetic. This has been clarified in the revised results and discussion sections upon the first mention of the acetylmimetic approach. Glutamate is not a good mimetic for acetyl-lysine because it has a negatively charged carboxylic acid side chain, which is not reflective of the charge or functional group of acetyl-lysine.

The K39E mutant was included to test the hypothesis that change in charge drives maximal functional effects. Given that lysine is positively charged, glutamine is neutral but polar like acetyl-lysine, and glutamate is negatively charged, the K39Q and K39E replacements produce a change of 1 and 2 charge units compared to WT, respectively. As predicted, K39E Cytc shows a more pronounced effect to activate COX activity compared to K39Q and WT, producing the trend: WT < K39Q < K39E Cytc (Figure 1G).

It should also be noted that we are not claiming that glutamine substitution is a perfect model for lysine acetylation; however, it replicates functional changes seen with acetylated Cytc and it makes possible to conduct experiments which are well controlled (e.g., 100% WT versus 100% acetylmimetic both for in vitro assays and cell culture experiments). Another advantage is that in contrast to glutamine substitution which will not change, the acetyl group could potentially be lost in certain assays (for example during the longer incubations with cytosolic cell extracts which contain deacetylase enzymes, such as the caspase assays).

As noted above, the K39 residue of Cytc demonstrates switch-like behavior whether the residue is positively charged vs. uncharged/negatively charged. While it seems to be the case that both K39Q and K39E Cytc demonstrate similar behavior for K39 of Cytc (but non-identical as shown in the example above for COX activity measurements shown in Figure 1G), this does not hold true for other residues of Cytc or other proteins in general. Anecdotally, we can share that for another acetylation site of Cytc, the glutamate mutant does not behave similarly to the glutamine mutant or the in vivo acetylated proteins (unpublished). Given the above explanation, we would not consider K39E to be an acetylmimetic.

Please also see reviewer 3 – comment 9 for our response.

2. Experimental controls on tissue treatments seem to be lacking. No information is provided to determine whether acetylation is statistically significant. I suggest the following:
 - a. Analyze by mass spectrometry and immunoblotting a larger number of tissue samples. For example, 12 tissue samples (2 from each hindlimb of three animals, 4 samples per animal) and report the results for all samples as supplementary information.
 - b. The work is partially funded by NIH, which requires sex to be considered as a biological variable in studies involving vertebrate animals. The sex of the animals has not been considered. At least, please indicate the sex of the animals used.
 - c. The process described in the methods section to induce ischemia involves keeping the tissue at 37 °C for 1 h. There is not reference provided in the manuscript to determine whether this is an acceptable protocol. Please, test the presence of acetylation as a function of time within the 1 h period and report the results.

We thank the reviewer for these suggestions. As requested, in the revised manuscript, we immunoblotted several control TA samples and ischemic TA samples as part of the time course experiment (see below). Additionally, we have analyzed 12 samples by mass spectrometry. There are 6 control TA muscle samples (3 male and 3 female) and 6 ischemic TA muscle samples (3 male and 3 female). All spectra have been uploaded to a depository. The accession code is PXD040915 and the DOI is 10.6019/PXD040915. Reviewer access may be found via the following information:

<https://www.ebi.ac.uk/pride/login>

Username: reviewer_pxd040915@ebi.ac.uk

Password: 2l4PJkZ

The proteome data will be publicly available upon acceptance of the manuscript. See reviewer 1 – comment 1 for our response regarding the results of the requested experiments.

Thank you for pointing out our oversight regarding sex as a biological variable. We have identified K39 acetylation in both male and female pigs. We have clarified this in the revised manuscript.

Regarding the ex vivo ischemia protocol, we have previously used it to induce brain ischemia ex vivo (PMID: 31570002). We have clarified this in the revised methods section. As requested, we tested the acetylation status as a function of time. We did this using immunoprecipitation experiments with TA muscle frozen after 0 (control), 15, 30, 45, and 60 minutes of ischemia. The TA muscle samples were lysed and immunoprecipitated for Cytc. The eluate off the IP beads was run on a gel and then probed using an anti-acetyl-lysine antibody to assess Cytc acetylation state. Interestingly, acetylation appears to be induced at 45 minutes of ischemia and potentially increases at 60 minutes of ischemia. We thank the reviewer for suggesting this experiment.

3. The labels and numbers in Figure 1a are barely visible. Please, increase font size and provide a better explanation in the figure of the results that need to be highlighted. Please, indicate how many mass spec. were obtained and how many immunoblotting gels were performed.

As requested, we have done the following: we have improved the resolution of Figure 1A as much as possible, we have edited the figure legend to better explain the results that need to be highlighted, we have indicated the number of replicates for each mass spectrum and immunoblot.

4. No clear justification is provided of why the K39Q mutant is a suitable mutation to represent acetylation.

Please see reviewer 4 – comment 1 for our response.

5. Figure 1d (not the inset) seems to show only three proteins: WT, K39E and K39R. If K39Q, which is the selected mutant is overlapping, please indicate so and indicate with which curve it is overlapping.

The reduced K39Q spectra is overlapping with the reduced K39R spectra. We have clarified this in the revised results section and the revised figure legend.

6. There is a significant difference between the caspase-3 activity of the K39Q mutant (90% decrease in activity) compared with the naturally acetylated version of the induced—ischemic sample (45% decrease activity). Although, the trend is in the right direction, these results imply that there are important differences between the mutant and the acetylated wildtype. The sentences in lines 109 and 100 need to indicate this significant difference to consider the suitability of the K39Q mutant.

The protein purified from ischemic muscle exists as a mixture of acetylated and unacetylated protein, whereas the K39Q protein mimics 100% acetylation. Therefore, we would expect the mimetic to show

more pronounced effects than the naturally acetylated protein. We have added language in the revised discussion section to discuss this.

7. Based on Figures 1 and 2, it is clear that K39 has important effects on Cyt_c function. However, the K39Q mutation does not seem to be a good mimetic. Data presented in Figures 1g, 2a and 2b, show that both mutants K39Q and K39E have similar effects, even though E is negatively charged, and Q is neutral as the acetylated K. Thus, references throughout the manuscript indicating that the K39Q mutant is a better mimetic are somewhat misleading, as well as the titles of the figure legends specifically highlighting this mutant.

Please see reviewer 4 – comment 1 for our response.

8. The symbols in Figures 1 and 2 (circle, inverted triangle, etc.) are very small, making it very hard for the reader to identify differences between them.

We have increased the size of these data point symbols (circle, square, triangle, inverted triangle, and diamond) by 25% in the revised manuscript.

9. The importance of the MD simulations does not seem to be clearly explained. The MD results show that the 20-30 loop in the K39 acetyl-lysine is more flexible than in the other structures. Is there an explanation for this result in relation to the claim of the manuscript?

See reviewer 2 – comment 2 for our response.

10. In relation to the crystallographic results, I suggest representing the potential changes in the structure of the backbone and side chains in the regions nearby K39Q and the acetylated K39 in the 3D space to facilitate the identification of the differences between them.

As requested, we have generated overlay images of rodent K39Q Cyt_c with rodent WT Cyt_c and porcine K39 acetylated Cyt_c with rodent WT Cyt_c within 3.5 Å of residue 39. These images are available in Supplemental Figure 1 and show that the K39 acetylated Cyt_c has greater variation at residue 39 compared to the K39Q Cyt_c, which we believe to be due to the acetyl-lysine side chain being slightly longer than the glutamine side chain. We have expanded the results section to comment on this.

11. The chemical shift perturbation data indicates that mutations in K39 by either E or Q led to structural changes (more pronounced for the less conservative mutation) in the regions close to amino acids 57 and 80. The mutation K39R is not perturbing the native structure significantly. However, this latter mutation results in modified function (e.g., Fig. 6b, c, d). Could the authors provide an explanation in the manuscript about this apparent contradiction?

Based on the NMR chemical shift perturbation data, the K39R mutation does not significantly perturb the overall structure, as the reviewer pointed out. The observed modified function for this variant (increased ATP levels and oxygen consumption rate) could be explained based on the proximity of residue 39 to the propionate heme groups, which can alter the heme moiety and redox properties of the metalloprotein. Additionally, arginine is different from lysine, specifically due to the different side chain functional groups and due to the positively charged side chain for arginine being delocalized. These differences are a potential cause of K39R acting as a hybrid between WT and K39Q for some assays. We have added language in the revised discussion section to explain this.

12. The differences in relaxation data between the WT protein and the K39Q mutant cannot be interpreted as presented because there are no errors associated to the measurements or at least the errors are not mentioned in the manuscript. By reading the methods section, it seems that the R1, R2 and het-NOE values

were obtained once. Some of the differences observed could be related to error measurement. NMR relaxation experiments must be acquired at least twice to be able to report error/deviation.

Following the reviewer recommendation, we have repeated the NMR relaxation experiments for the WT and K39Q variants to corroborate the observed differences. We have updated revised Figure 5 and revised Supplemental Figure 8 to include the data of both replicates.

13. The results on the effect of K39Q in the increase of ATP levels compared to WT and the basal oxygen consumption rate are similar to the other mutant K39R and K39E. These results are analogous to data shown in Figure 1 and 2, where we can conclude that K39 is an important amino acid controlling these functions, but the K39Q is as good mimetic as K39E and K39R in some instances.

Please see reviewer 3 – comment 9, reviewer 4 – comment 1, and reviewer 4 – comment 11 for our response.

There appears to be a switch-like behavior for Cytc at residue 39 whether the residue is positively charged or uncharged but polar/negatively charged. Indeed, K39R does demonstrate irregular behavior for the ATP assay, which we have commented on in the revised discussion section.

REVIEWERS' COMMENTS

Reviewer #1 (Remarks to the Author):

The authors have given thoughtful and helpful responses to my comments, and their subsequent edits have improved the manuscript. I have no further questions or suggestions prior to publication.

Reviewer #2 (Remarks to the Author):

The authors have addressed all my concerns

Reviewer #3 (Remarks to the Author):

The comments from the previous review have been addressed in sufficient detail. The sections on X-ray crystallography look to be in good shape now.

The comparison with the prior phosphorylated crystal structures shows interesting results, perhaps a future direction? Reasons other than the level of chemical perturbation that might be causing this effect.

Reviewer #4 (Remarks to the Author):

Based on the results presented in this manuscript, several reviewers point out that the Lys to Gln mutation does not seem to be a better mimetic of Acetyl-Lys for CytC than other mutations, despite the Lys/Gln mutation perhaps working well in other systems. I would have preferred that the authors stated this fact; however, overall, the authors have addressed most of the concerns and thus I recommend publication of the revised version.

Reply to the reviewers' critiques:

Reviewer #1 (Remarks to the Author): The authors have given thoughtful and helpful responses to my comments, and their subsequent edits have improved the manuscript. I have no further questions or suggestions prior to publication.

Response: We thank the reviewer for their positive assessment.

Reviewer #2 (Remarks to the Author): The authors have addressed all my concerns.

Response: We thank the reviewer for their positive assessment.

Reviewer #3 (Remarks to the Author): The comments from the previous review have been addressed in sufficient detail. The sections on X-ray crystallography look to be in good shape now. The comparison with the prior phosphorylated crystal structures shows interesting results, perhaps a future direction? Reasons other than the level of chemical perturbation that might be causing this effect.

Response: We thank the reviewer for their positive assessment and suggestion for future analyses.

Reviewer #4 (Remarks to the Author): Based on the results presented in this manuscript, several reviewers point out that the Lys to Gln mutation does not seem to be a better mimetic of Acetyl-Lys for Cyt_c than other mutations, despite the Lys/Gln mutation perhaps working well in other systems. I would have preferred that the authors stated this fact; however, overall, the authors have addressed most of the concerns and thus I recommend publication of the revised version.

Response: We thank the reviewer for their positive assessment. To address the reviewer's point regarding the acetylmimetic substitution we have added the following sentence in the discussion section: Interestingly, in some experiments the glutamate substituted Cyt_c produced results comparable to those obtained with the acetylmimetic glutamine, suggesting that other modifications of this residue, including the introduction of a negative charge via the glutamate side chain, on the highly evolutionary optimized Cyt_c molecule can cause similar functional effects.